# Multiparametric senescent cell phenotyping reveals targets of senolytic therapy in the aged murine skeleton

Madison L. Doolittle [1,2], Dominik Saul [1,2,3], Japneet Kaur[1,2], Jennifer L. Rowsey[1,2], Stephanie J. Vos[1,2], Kevin D. Pavelko [4], Joshua N. Farr [1,2], David G. Monroe [1,2] & Sundeep Khosla [1,2] ✉

Senescence drives organismal aging, yet the deep characterization of senescent cells in vivo remains incomplete. Here, we apply mass cytometry by time-of-flight using carefully validated antibodies to analyze senescent cells at single-cell resolution. We use multiple criteria to identify senescent mesenchymal cells that are growth-arrested and resistant to apoptosis. These p16 + Ki67-BCL-2+ cells are highly enriched for senescence-associated secretory phenotype and DNA damage markers, are strongly associated with age, and their percentages are increased in late osteoblasts/osteocytes and CD24[high] osteolineage cells. Moreover, both late osteoblasts/osteocytes and CD24[high] osteolineage cells are robustly cleared by genetic and pharmacologic senolytic therapies in aged mice. Following isolation, CD24+ skeletal cells exhibit growth arrest, senescence-associated β-galactosidase positivity, and impaired osteogenesis in vitro. These studies thus provide an approach using multiplexed protein profiling to define senescent mesenchymal cells in vivo and identify specific skeletal cell populations cleared by senolytics.

Cellular senescence is a state of proliferative arrest that occurs due to the accumulation of DNA damage and cellular stress[1–4]. This is distinct from quiescence, as senescent cells can acquire a senescence-associated secretory phenotype (SASP), consisting of pro-inflammatory factors that have detrimental effects on cell and tissue function both locally and systemically[5]. Senescent cells accumulate in the bone microenvironment with age[6], and clearance of senescent cells in old mice alleviates multiple age-related morbidities[7–12], increases lifespan[13–15], and preserves bone microarchitecture and strength[16]. Thus, senescent cell clearance represents a therapeutic approach to delay or alleviate age-related diseases.

The direct identification of cells undergoing senescence in vivo would allow for biological age phenotyping, determination of senolytic efficacy, and identification of novel senolytic targets; however, multiple technical obstacles prevent this process. The variable abundance and rarity of tissue-resident senescent cells[17] requires large sample sizes to provide sufficient signal and statistical power[18]. Moreover, established markers for senescence (e.g., p16[Ink4a], p21[cip1]) are expressed intracellularly and at low levels, typically restricting senescence phenotyping to whole-tissue RNA analyses. This method of bulk phenotyping unfortunately lacks the resolution to determine if these cell-cycle proteins are upregulated in the same cells exhibiting a SASP and growth arrest, blurring the distinction between senescence and systemic inflammation. Compounded by their expected heterogeneity[19], it remains difficult to comprehensively study senescent cells in vivo without increased cellular resolution.

Single-cell RNA sequencing technologies have made significant technological advances, yet studies using this approach to investigate senescence present their own challenges. Though overcome by meticulous PCR primer design, the genes encoding both p16[Ink4a] (*Cdkn2a*)

---

[1]Division of Endocrinology, Diabetes and Metabolism, Mayo Clinic, Rochester, MN 55905, USA. [2]Robert and Arlene Kogod Center on Aging, Mayo Clinic, Rochester, MN 55905, USA. [3]Department for Trauma and Reconstructive Surgery, BG Clinic, University of Tübingen, Tübingen, Germany. [4]Department of Immunology, Mayo Clinic, Rochester, MN 55905, USA. ✉e-mail: khosla.sundeep@mayo.edu

and p21cip1 (*Cdkn1a*) proteins generate multiple transcript variants that are challenging to segregate through standard single-cell transcriptomic libraries; *Cdkn2a* also encodes p19Arf (p14Arf in humans), a protein with separate functions and opposing effects on senescence to p16Ink4a [20–22], and *Cdkn1a* has multiple variants that associate with aging to varying degrees[23]. Moreover, discrepancies between mRNA and protein expression have been observed for key senescence markers, including p16Ink4a [24], which is compounded by the effects of senescence on translation[25] and proteostasis[26]. Therefore, single-cell techniques leveraging multiplexed protein profiling to assess p16 or p21 proteins, as well as other properties of senescent cells (e.g., the SASP, growth arrest, apoptosis resistance), would potentially be a significant advance for studying the fundamental biology of senescence.

In the present study we leverage the power of mass cytometry by time-of-flight (CyTOF)[27] to define, at the single-cell level, mesenchymal senescent cells in vivo. Specifically, rather than relying principally on p16 or p21 expression, we used multiple criteria to define senescent cells. Moreover, given the inherent biological variability of senescent cells with aging[28], we used a large sample size of mice across all experiments (*n* = 88), assessed senescent cell burden in established skeletal cell populations, and tested their susceptibility to senolytic clearance through either genetic or pharmacologic approaches. Findings by CyTOF were supported by single cell RNA-sequencing (scRNA-seq) and in vitro phenotyping. Collectively, our studies establish a robust approach to define senescent mesenchymal cells in vivo using CyTOF, provide a detailed map of aging- and senolytic-induced alterations in mesenchymal cells in the murine bone microenvironment, and identify specific osteolineage populations that are highly inflammatory, senescent using rigorous criteria, and cleared by senolytics.

## Results

### Development and validation of a senescence CyTOF antibody panel

We constructed and validated a comprehensive CyTOF antibody panel to include markers for both cell identity and senescent phenotype (Table 1). A defining characteristic of senescent cells is expression of cell cycle inhibitors, in particular p16 or p21[29], so we carefully validated antibodies to these proteins for use in CyTOF. Due to concerns regarding the specificity of antibodies to mouse p16, we tested 3 separate commercial antibodies using an in vitro workflow (Fig. 1A–C). Mouse p16 was expressed in human U2OS cells through vector transfection alongside an empty vector control. While some antibodies demonstrated high background and limited positive signal by CyTOF, one antibody in particular provided an excellent signal-to-noise ratio (Fig. 1C) and was therefore selected for the panel. Notably, this antibody has been used in other senescence studies for flow or mass cytometry[30,31]. A similar process was followed for antibodies targeting several additional antigens, including p21 (Supplementary Fig. 1A–D).

To validate our senescence/SASP antibodies and confirm our technical ability to process, stain, and detect senescent cells, our panel was tested on etoposide-induced senescent mouse bone marrow stromal cells (BMSCs) (Fig. 1D). Compared to non-senescent BMSCs, senescent BMSCs demonstrated an upregulation of cells positive for p16, p21, and a majority of our senescence/SASP markers by CyTOF (Fig. 1E, F).

We also optimized our single-cell suspension collection protocol from mouse bone and marrow: as shown by others[32,33], marrow digestion was optimized to release high yields of non-hematopoietic cells from marrow stroma (Supplementary Fig. 2A, B). Additionally, bone tissue digestion was optimized to ensure release of both early (Runx2+, Osterix+) and late (ALPL+, Osteocalcin+) osteolineage cells (Supplementary Fig. 2C–F), which also assisted in validating our osteoblast lineage cell antibodies[34,35]. This protocol is similar to workflows used by other laboratories for single-cell analysis of the mesenchymal bone microenvironment[35–37].

We next applied our CyTOF antibody panel to freshly isolated bone and marrow mesenchymal cells from *INK-ATTAC* mice: a transgenic model with an inducible caspase 8 cassette driven by the p16Ink4a promoter, allowing for selective clearance of senescent cells through treatment with AP20187 (AP)[12]. Using this approach, we could robustly compare cellular alterations resulting from both aging and senolytic clearance within the same mouse strain; accordingly, we collected a total of 40 mice in these groups (*n* = 15 "Young" [6-month], *n* = 12 "Old" [24-month] + vehicle, and *n* = 13 Old [24-month] + AP). We applied our antibody panel to single-cell suspensions of non-immune (Lin- [CD5, CD45R (B220), CD11b, Gr-1 (Ly-6G/C), Ly6B.2 and Ter-119]) and non-hematopoietic (CD45-) mesenchymal skeletal cells (Fig. 1G, see Supplementary Fig. 2G for gating strategy) using our optimized protocol for harvesting mesenchymal lineage cells from both bone and marrow.

Using identically processed cells from p16 knock-out mice[38] as a negative control (Fig. 1H), we identified p16+ mesenchymal cells from both young and old mice (Fig. 1I). We emphasize that our identification of p16+ cells in vivo by CyTOF utilized an antibody that specifically detected the mouse p16 protein in human cells expressing the p16Ink4a construct (Fig. 1C), demonstrated an increased signal for p16+ cells in vitro following etoposide treatment (Fig. 1E), and was thresholded on p16 knock-out cells (Fig. 1H). It is also important to note that the validation of this antibody at this point is only for single-cell CyTOF and other uses (e.g., immunohistochemistry, etc.) would require independent validation of this antibody for that specific purpose. Using this antibody, we found that p16+ cells were more abundant with age, expanding from 2.81% ± 1.32% to 7.60% ± 5.60% (*P* = 0.004) of the total cell population from 6 to 24 months of age, respectively (Fig. 1I). In contrast, the prevalence of p21+ mesenchymal cells did not increase with age in the bone microenvironment (Fig. 1J; 5.21% ± 3.27% in young, 3.71% ± 2.96% in old [*P* = 0.241]).

### Single-cell specification of senescent mesenchymal cells in bone and marrow

Although found to be associated with age, p16 positivity can also be a characteristic of non-senescent cells[39–42], and thus in isolation may not identify a pure population of truly senescent cells. Therefore, to define additional markers uniquely associated with age-induced senescent cells, we performed multidimensional clustering of p16+ mesenchymal cells from young and aged mice using all markers in our senescence panel (Fig. 2A, Supplementary Fig. 3A). We identified considerable cellular heterogeneity, with p16+ cells segregating into 6 unique clusters defined by expression of BCL-2, Ki67, γH2A-X, or other inflammatory markers, along with an unlabeled cluster (Fig. 2B). The BCL-2+ and BCL-2+/γH2A-X+ populations showed an increased trend with age, while the Ki67+ populations were reduced (Fig. 2C). Similar to recent findings using highly sensitive p16Ink4a-reporter mice (INKBRITE)[39], we found an inverse relationship between Ki67 and p16 expression (Supplementary Fig. 3B), which also revealed that clusters both high in p16 and low in Ki67 were defined by expression of BCL-2, a major regulator of apoptosis resistance[43] (Fig. 2D). Figure 2D also clearly demonstrates that as p16 expression increased across the clusters (red line), Ki67 expression concurrently decreased (blue line). We then performed manual gating on our samples and found that the BCL2+ subset of p16+ cells was inherently reduced in both Ki67 expression and Ki67+ cells (Fig. 2E, F), with nearly 98% of p16 + BCL2+ cells being Ki67- (Supplementary Fig. 3C). Across aging, we found that BCL-2 was the only factor clearly upregulated in old versus young p16+ cells (1.8-fold, adj. p = 0.0012; Fig. 2G); this finding is entirely consistent with extensive literature demonstrating upregulation of BCL-2 anti-apoptosis pathways in senescent cells[44–46].

Based on this data, we defined a mesenchymal senescent cell population at the single cell level as being p16+/Ki67-/BCL-2+ (hereafter referred to as "p16KB" cells), which constituted 17.44% of p16+ cells (Fig. 2H). To support this, we found that p16KB cells exhibited

**Table 1 | CyTOF Antibody Panel and Validations**

| Identification | Marker | Metal conjugation | Commercial validation | | In-house validation | | |
|---|---|---|---|---|---|---|---|
| | | | CyTOF | Flow cytometry | Expression construct | Etoposide-induced senescence | Digested bone samples |
| Hematopoietic | CD45 | 89Y | ✓ | | | | |
| Perivascular | CD146 | 141Pr | ✓ | | | | |
| Bone Marrow Stromal Cells (BMSCs) | LeptinR | 146Nd | ✓ | | | | |
| | Nestin | 168Er | | | | | |
| | Sca-1 | 169Tm | ✓ | | | | |
| | CD24 | 142Nd | | ✓ | | | |
| | CD140a / PDGFRα | 148Nd | | ✓ | | | |
| | SDF-1 / CXCL12 | 144Nd | ✓ | | | | |
| | Itga11 / OsteolectinR | 156Gd | | | | | |
| | CD200 | 152Sm | | ✓ | | | |
| | CD29 | 158Gd | | ✓ | | | |
| Osteoblast | Runx2 | 147Sm | | ✓ | | | ✓ |
| | SP7 | 151Eu | | | | | ✓ |
| | ALPL | 154Sm | ✓ | | | | ✓ |
| | OCN | 111 Cd | | | ✓ | | ✓ |
| Osteocyte | Dmp1 | 143Nd | | | ✓ | | |
| | Podoplanin | 174Yb | | ✓ | ✓ | | |
| | Sclerostin | 166Er | | ✓ | ✓ | | |
| Adipocyte | PPARγ | 153Eu | | ✓ | | | |
| | Adiponectin | 170Er | | ✓ | | | |
| Transgene | FLAG | 164Dy | | ✓ | ✓ | ✓ | |
| SASP | MCP-1 | 149Sm | | ✓ | | ✓ | |
| | TNFα | 162Dy | ✓ | | | ✓ | |
| | PAI-1 | 160Gd | | ✓ | | ✓ | |
| | IL-6 | 167Er | ✓ | | | ✓ | |
| | IL-1α | 171Yb | | | | ✓ | |
| | IL-1β | 159Tb | | ✓ | | ✓ | |
| | CXCL1 | 175Lu | | | | ✓ | |
| | pNFkB | 110 Cd | | ✓ | | ✓ | |
| Senescence | CENP-B | 145Nd | | | | ✓ | |
| | p21 | 176Yb | | ✓ | ✓ | ✓ | |
| | p16 | 155Gd | | ✓ | ✓ | ✓ | |
| | p53 | 116 Cd | | ✓ | | ✓ | |
| DNA Damage | yH2A-X | 173Yb | | ✓ | | ✓ | |
| | pATM | 114 Cd | | ✓ | | ✓ | |
| Anti-Apoptosis | Bcl-2 | 112 Cd | | | ✓ | ✓ | |
| Proliferation | Ki67 | 106 Cd | ✓ | | | | |

Antibodies used in the CyTOF panel were commercially validated for CyTOF/flow cytometry and/or validated in-house using several different approaches. Checkmark indicates completed validation.

upregulated expression of numerous markers for senescence (p21), SASP (IL-1α, IL1β, pNFκB, CXCL1) and DNA damage (pATM) (Fig. 2I), though it should be noted that p16KB SASP signatures exhibited inter-animal heterogeneity. Importantly, however, p16KB cells exhibited higher expression levels of a majority of senescence and SASP markers than parental p16+ cells (Supplementary Fig. 3D). Moreover, p16KB bone and marrow cells were highly age-associated, making up only <0.2% of all cells in young mice, yet with a fold-change of 6.8 across aging (Fig. 2J) (as compared to total p16+ cells: 2.81% in young, 2.7-fold with age, Fig. 1I).

We next performed a deeper analysis of p16KB cells through single-cell RNA-sequencing of bone and marrow cells following enrichment for mesenchymal cells as for our CyTOF samples. We were able to identify *Cdkn2a*+ (p16[Ink4a]+) cells, which contained mutually exclusive *Mki67*+ or *Bcl2*+ subpopulations (Supplementary Fig. 3E, F). The p16KB (*Cdkn2a*+ *Mk67-Bcl2*+) population (Cluster 4) demonstrated robust co-expression of other well-characterized anti-apoptosis genes, including *Bcl2l2* (BCL-W)[47], *Mcl1*[48,49]*, Xiap*[50]*, Birc6*[51], as well as *Mapk14*, which encodes the major subunit (p38α) in p38MAPK, a robust regulator of senescence and SASP[52–54] (Supplementary Fig. 3G). To further investigate the SASP in these cells, we performed CellChat analysis and found that the p16KB cell cluster exhibited the highest levels of outgoing secretory signals of any *Cdkn2a*+ population (Supplementary Fig. 3H–J). We then applied our recently established senescence gene panel ("SenMayo")[55] and indeed found the highest enrichment for senescence/SASP genes in p16KB cells (Supplementary Fig. 3K, L).

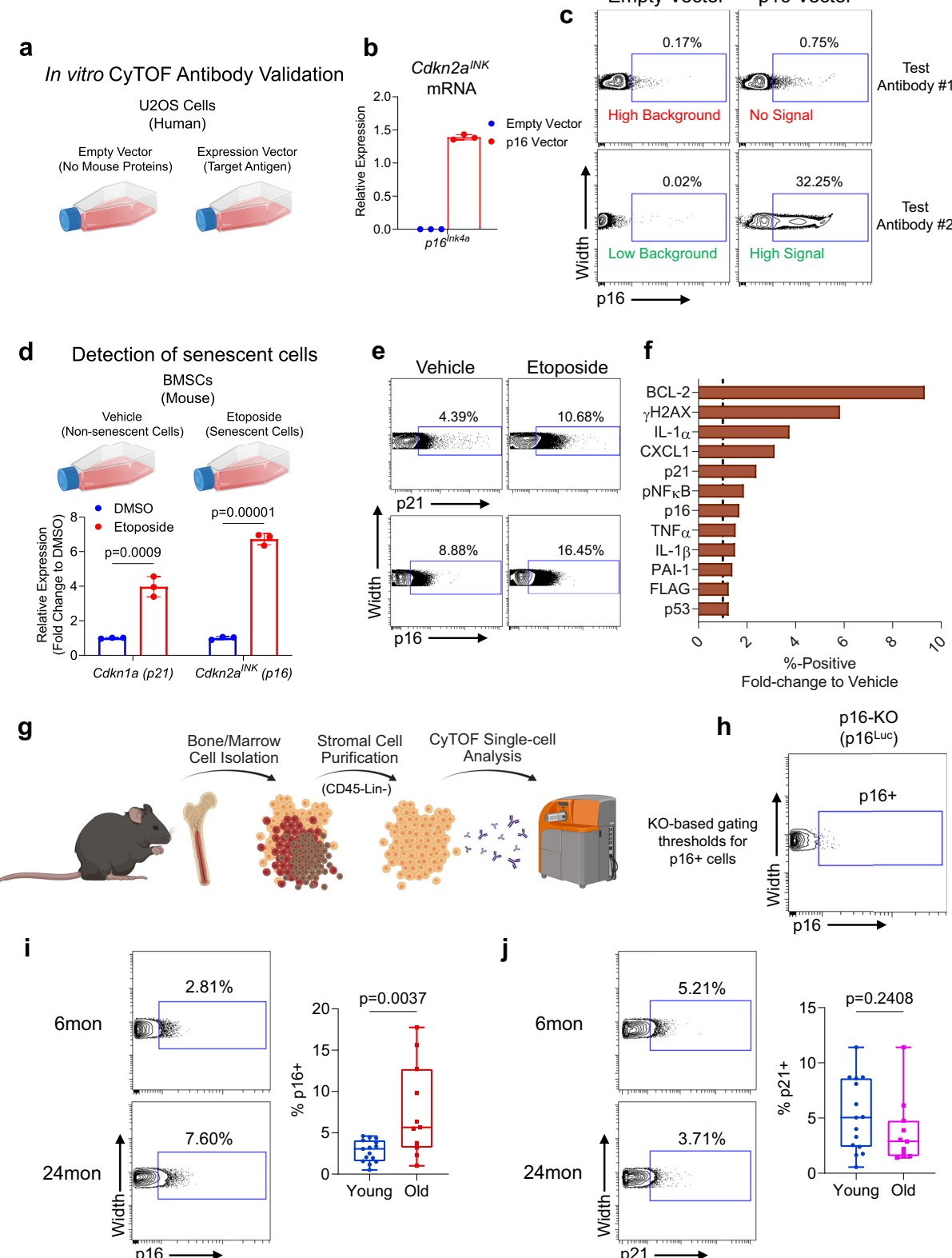

Similar to p16, BCL-2+ subsetting of p21+ cells identified a population with reduced Ki67 positivity compared to total p21+ cells (Supplementary Fig. 4A). Analogous p21+/Ki67-/BCL-2+ ("p21KB cells") were identified (Supplementary Fig. 4B), which displayed a robust SASP and DNA damage expression profile (Supplementary Fig. 4C). Interestingly, despite no age-related increase in total p21+ cells, p21KB cells did increase significantly, albeit modestly, in old mice

(Supplementary Fig. 4D), thus identifying an age-associated p21+ subpopulation. However, p16KB cells greatly outnumbered p21KB cells in old mice and exhibited a more dramatic upregulation with age (Supplementary Fig. 4E). Upon further investigation, we found minimal overlap between p16KB and p21KB cells (0.04% of total cells in young, 0.10% in old; Supplementary Fig. 4F). To test the validity of both p16KB and p21KB results, we replicated this CyTOF analysis in an additional

**Fig. 1 | Validation of antibodies and detection of senescent cells by CyTOF.**
**a** Experimental workflow of single mouse protein expression in U2OS cells for the testing of CyTOF antibodies; **b** qPCR analysis confirming upregulated mouse p16[Ink4a] mRNA (*Cdkn2a*) after expression vector transfection (*n* = 3 independently run experiments; mean ± SD); **c** CyTOF dot plots of expression samples testing different p16 antibodies, demonstrating outcomes of both failed (antibody #1) and successful (antibody #2) tests; **d** Schematic of testing senescence-specific CyTOF antibodies using etoposide-treated mouse BMSCs with qPCR confirmation of upregulated p16[Ink4a] (*Cdkn2a*[INK]) and p21[Cip1] (*Cdkn1a*) transcripts (*n* = 3 independently run experiments; mean ± SD); **e** CyTOF plots of p16 and p21 protein expression, demonstrating increased percent-positive cells with etoposide treatment; **f** Fold-change of percent-positive values for each of the senescence panel markers in etoposide-treated cells, where dotted line represents values from vehicle. **g** Schematic of bone and marrow mesenchymal cell isolation and CyTOF workflow; **h** Gating strategy for p16+ cells using similarly processed cells from p16-null mice as a negative control (**i**) Quantification of %p16+ and (**j**) %p21+ cells in young and old mice (*n* = 15 young and *n* = 12 old biologically independent animals). Schematics in (**a**, **d**, **g**) were generated using BioRender. Box plots show median and interquartile range with error bars representing minimum and maximum values. (**d**) Multiple two-sided t tests with Holm-Sidak Correction. (**i**, **j**) Two-sided Mann-Whitney test. Source data are provided as a Source Data file.

---

cohort of young (6-month) and aged (24-month) C57BL/6N mice (wild-type, rather than *INK-ATTAC*) (Supplementary Fig. 4G). We found our observations to be robustly conserved, revealing similar age-associated patterns for total p16+ and p21+ cells, with BCL-2 subsetting identifying growth-arrested and age-associated subpopulations for each (Supplementary Fig. 4H–J).

Collectively, using single-cell mass cytometry, we thus define p16KB and p21KB mesenchymal cells as being senescent, as these populations fulfill virtually all of the required criteria for senescent cells[56]: increased p16 or p21 protein expression, growth arrest, upregulation of anti-apoptotic pathways, upregulation of SASP and DNA damage markers, and a relatively low abundance in youth with a marked increase with aging. Moreover, we note that in addition to our use of carefully validated antibodies in the CyTOF analysis, our orthogonal approach to identify p16KB and p21KB cells does not rely solely on the specificity of any single antibody, but rather uses a combinatorial approach, making it extremely unlikely that the p16KB or p21KB senescent cells represent a false positive signal.

**Reconstruction of bone and marrow mesenchymal populations through CyTOF**

Next, in order to assess which mesenchymal skeletal cell populations harbored senescent cells with age, we established our bone-resident clusters. Using t-SNE visualization and FlowSOM cell clustering, we identified 11 populations of BMSCs and differentiated cell types with expression profiles consistent with the established literature (Fig. 3A, B; see Table 2 for a detailed description of defining markers for each population). These included LeptinR+ BMSCs[57], Sca-1+/PDGFRα+ BMSCs (also termed PαS cells)[58–60], perivascular BMSCs[61,62], and Nestin + pericytes[63]. Other characteristics of these cells have been established by recent studies, such as expression of CXCL12[64] and OsteolectinR (*Itga11*)[65] in LeptinR+ BMSCs, Adiponectin expression in perivascular and stromal cells[66], and PDPN (Podoplanin) expression in Sca-1/PDGFRα + BMSCs[60,67]. Additional clusters representing committed cell types included early osteoblasts (Runx2+/Osterix+)[68,69], alkaline phosphatase (ALPL)+ osteolineage cells (Runx2+/ALPL+)[70,71], late osteoblasts/osteocytes (Runx2+/Osterix+/Sclerostin+; note that our cell isolation protocol included a collagenase digestion thereby releasing at least a subset of osteocytic cells)[72] and pre-adipocytes (Pparg+)[73,74]. Other populations represented less well-defined populations, including CD24[high/low] osteolineage (CD24+/Runx2+/Osterix+) and CD24+/Osterix+ clusters. Note that the latter cells emerged as a distinct cluster and had lower expression of Runx2 than the CD24[high/low] osteolineage (CD24+/Runx2+/Osterix+) cells (Fig. 3B), yet had strong Osterix positivity within a large portion of this cluster (Supplementary Fig. 5A, B).

CD24 has previously been linked to osteogenesis[75], while conversely also shown to mark pluripotent BMSCs[76,77], particularly when co-expressed with Sca-1[78] or CD200; however, these CD24 clusters lacked co-expression of these stem-like markers (Supplementary Fig. 5A, B) indicating a distinct, more committed osteoblast lineage population. Furthermore, histological staining of CD24 in femur sections from *Runx2*-TdTomato reporter mice demonstrated co-expression in cells at the trabecular bone surface, exhibiting morphology consistent with osteoblasts and/or bone lining cells (Supplementary Fig. 5C). CD24 staining was also evident in marrow cells lacking Runx2 expression, likely B cells[79], but CD24 was not expressed in osteocytes. To further investigate the CD24 subpopulations (CD24+Runx2+Osterix+ vs CD24+Osterix+), we performed CITE-Seq with the CD24 antibody on mesenchymal bone and marrow cells, prepared in a similar manner to our CyTOF experiments (Supplementary Fig. 5D). We found that CD24 marked a population that co-expressed *Runx2* and Sp7 (Osterix) and cells expressing all three markers designated a distinct osteogenic population (Supplementary Fig. 5D). We identified CD24+*Runx2*+Sp7+ and CD24+Sp7+ populations, finding that the CD24+Sp7+ subpopulation co-expressed late osteogenic markers (e.g., *Alpl*, *Bglap*) (Supplementary Fig. 5E). There was less segregation between these populations than what was observed at the protein level by CyTOF, which suggests possible post-translational degradation of RUNX2 that generates a distinct CD24+Osterix+ population. The gene expression signature of CD24+ cells appeared to reflect more mature osteolineage populations, as all CD24+ cells were enriched for late osteogenic (*Bglap, Bglap2, Postn*) markers, while being downregulated in stem-like markers (*Lepr, Adipoq, Cxcl12*) (Supplementary Fig. 5F). These changes were even more marked in CD24 osteolineage cells (CD24+*Runx2*+Sp7+) (Supplementary Fig. 5G).

We next evaluated our CyTOF panel for its ability to delineate the various stages of mesenchymal cell differentiation. Diffusion mapping displayed perivascular BMSCs and Nestin+ pericytes on one end of the lineage continuum, and Sca-1/PDGFRα+BMSCs on another, converging to form several committed osteogenic clusters (Supplementary Fig. 6A). This is consistent with many studies that demonstrate the multipotency of these populations and indicate multiple sources of osteolineage cells[80]. To determine directionality, we performed pseudotime analysis and trajectory inference, seeking to recapitulate the dynamics of in vivo cell differentiation. The progression of our clusters was consistent with our current understanding of biological mesenchymal differentiation of skeletal stem cells (Supplementary Fig. 6B, C): BMSC clusters are present early, then leading to early osteoblasts, ALPL+ osteolineage, and ultimately late osteoblast/osteocyte clusters. Interestingly, it appeared that the CD24+ clusters formed their own bifurcation from the typical differentiation of BMSCs to late osteoblasts and osteocytes (Supplementary Fig. 6A) and the CD24+ populations appeared to be middle (CD24[low/high] osteolineage [CD24+/Runx2+/Osterix+]) to later (CD24+/Osterix+) in the pseudotime progression (Supplementary Fig. 6B, C), consistent with their lack of expression of stem cell markers such as Sca-1 or CD200 (Supplementary Fig. 5A, B).

To further validate our CyTOF clusters, we performed single-cell RNA-sequencing (scRNA-seq) on a similarly prepared sample of bone and marrow cells from aged mice (Supplementary Fig. 7A). After multidimensional analysis, we found that skeletal cell populations identified through single-cell transcriptomics were entirely consistent with our CyTOF clusters (Supplementary Fig. 7B). Specifically, the BMSC populations (Sca-1/Pdgfra+, Perivascular, and Nestin+ Pericytes)

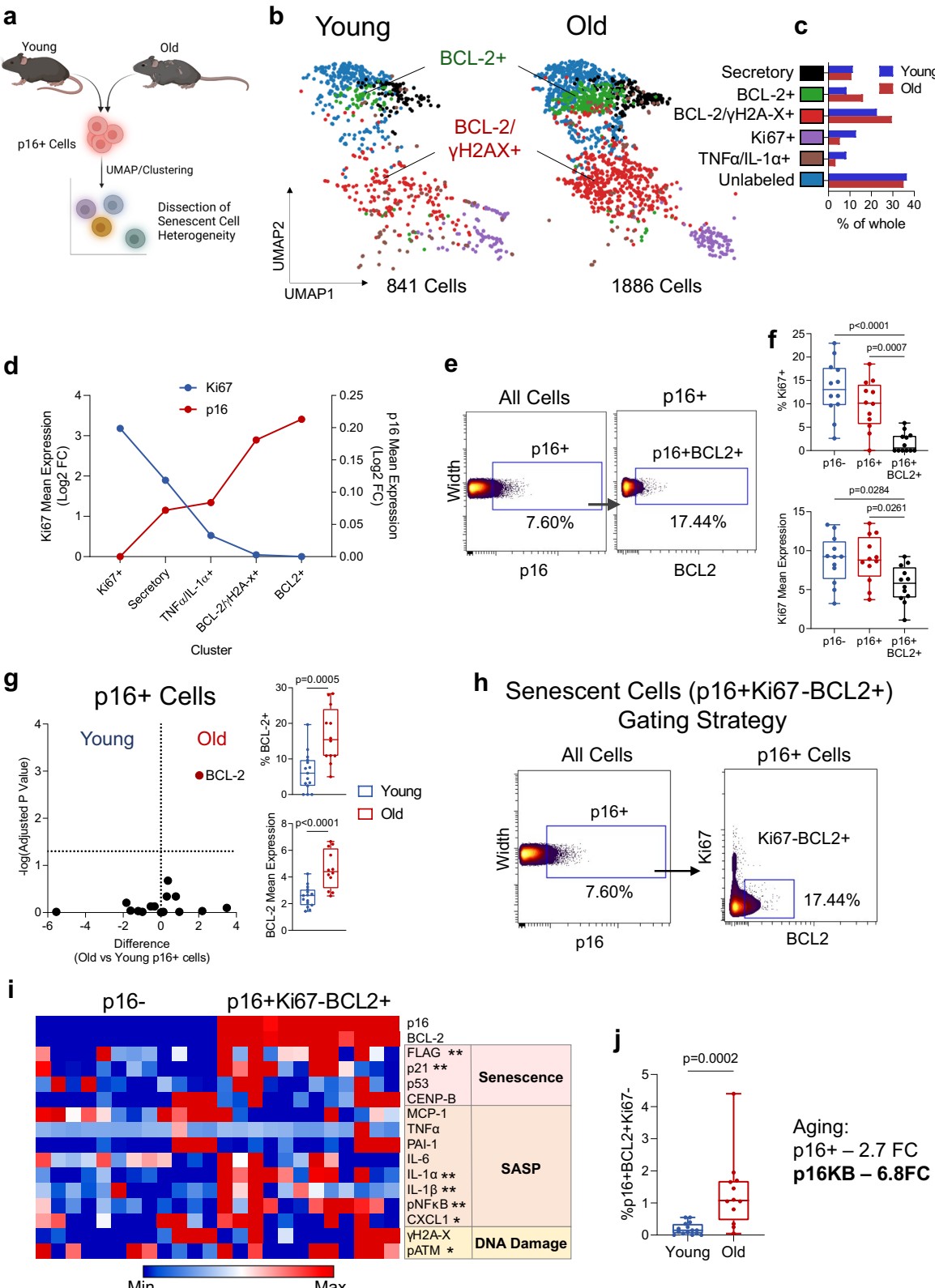

and CD24 osteolineage populations (*Cd24a*, *Runx2*) appeared well conserved, in addition to the distribution of expression for certain markers (e.g. *Cxcl12*) (Supplementary Fig. 7C). However, *Sost* (Sclerostin) was expressed at very low/undetectable levels, which has been observed by others, even within osteocyte-enriched samples[81]. Besides these limitations by scRNA-seq, the overall population structure was comparable to that which we observed using CyTOF, adding

confidence that this proteomic approach provides accurate interpretations of the bone microenvironment.

## Mature osteolineage populations harbor age-induced senescent cells

To identify mesenchymal populations affected by aging, we next examined senescence-specific effects, focusing on p16KB senescent

**Fig. 2 | BCL-2 expression defines p16+ cells with senescent characteristics.**
**a** Schematic of multidimensional p16+ senescent cell analysis workflow; **b** UMAP visualization and FlowSOM clustering of pooled p16+ cells from young ($n = 15$) and old ($n = 12$) mice with **c** bar graphs indicating percent-of-whole cluster abundance changes. **d** FlowSOM clusters ranked by descending Ki67 mean expression plotted against p16 mean expression (Log2 Fold-Change). **e** Gating strategy for p16 + BCL2+ cells and **f** quantification of %Ki67+ and Ki67 mean expression alongside p16- and total p16+ cells. **g** Volcano plots of age-related changes in mean senescence marker expression within p16+ cells and quantification of % BCL-2+ cells and mean BCL-2 expression within all p16+ cells with age; **h** Gating strategy for "p16KB" cells;

**i** Heatmap representation of protein expression between p16KB cells and non-senescent p16- cells, with asterisks indicating significance (*$p < 0.05$, **$p < 0.01$). Columns within each population represent a biological replicate ($n = 12$ Old). **j** Quantification of p16KB cells with age and numerical comparison to total p16+ cells ($n = 15$ young and $n = 12$ old biologically independent animals). Schematic in (**a**) was generated using BioRender. Box plots show median and interquartile range with error bars representing minimum and maximum values. (**f, g** [volcano plot]) Multiple two-sided t tests with Holm-Sidak Correction. (**g, i, j**) Two-sided Mann–Whitney test. Source data are provided as a Source Data file.

cells, as we have previously demonstrated that clearing p16+ cells in *INK-ATTAC* mice prevents age-related bone loss, reduces bone resorption, and increases bone formation[16]. We found that multiple clusters exhibited upregulation of p16 and/or BCL-2 expression with age (Fig. 3C), with many of these clusters exhibiting an expected increase in the percentage of senescent p16KB cells with age (Fig. 3D).

We next tested if the above populations were those that are cleared in aged *INK-ATTAC* mice (Fig. 3E; Supplementary Fig. 8A). Upon treatment with AP, cells in the late osteoblast/osteocyte cluster were markedly reduced (Fig. 3F, G), which is consistent with our previous data demonstrating a central role of osteocyte senescence with age[6,16,82–84]. In addition to this confirmatory result, we also uncovered that the CD24[high] osteolineage population was markedly reduced following AP treatment (Fig. 3F, G), along with a modest reduction in CD24+Osterix+ cells. Due to the substantial effect of p16+ cell clearance on CD24+ cell clusters (Fig. 3H), we further investigated what might underlie this effect. To do so, we leveraged the *INK-ATTAC* transgene (FLAG-tagged caspase 8 driven by an endogenous p16 promoter fragment) to monitor p16[INK4a] transcriptional activation[12,13]. When overlaying the expression of p16 along with FLAG on our clustered populations (Fig. 3I), we found concordance between the endogenous p16 protein and the caspase 8/FLAG protein produced by the transgene promoter fragment. Moreover, the 2 populations most robustly cleared by AP in the *INK-ATTAC* mice – the CD24[high] osteolineage cells and the late osteoblasts/ osteocytes (Fig. 3F, G) – exhibited both high p16 and caspase 8/FLAG expression (arrows in Fig. 3I). There was modest clearance of CD24+/Osterix+ cells (Fig. 3G), and although these cells did not have particularly high p16 and caspase 8/FLAG levels, clearly those levels were sufficient to reduce their numbers following AP treatment.

To test if CD24+ cells were directly killed by transgenic p16+ cell clearance, we isolated skeletal mesenchymal cells from 24-month-old *INK-ATTAC* mice in a similar manner to our CyTOF preparations and treated the cells with AP20187 in vitro for 24 h. We found increased apoptosis in CD24+ cells following AP20187 treatment, as demonstrated by increased % Annexin V+ cells by flow cytometry, with no such change in CD24- cells (Supplementary Fig. 8B), providing further evidence that this population is targeted in the bone microenvironment of *INK-ATTAC* mice. However, despite these findings demonstrating the pro-apoptotic effects of AP20187 on cell populations enriched for p16+ cells, it is possible the robust in vivo reduction of late osteoblastic/osteocytic cells observed by CyTOF may also be driven by additional non-apoptotic routes, including that treatment with AP20187 resulted in blockade of terminal differentiation of early osteoblasts, and additional studies are needed to address this possibility.

We also acknowledge that there do appear to be some populations (e.g., early osteoblasts) where sub-populations of cells have high p16/caspase 8/FLAG expression, but do not appear to be cleared by AP (Fig. 3H, I). The reasons for this are unclear at present, but it is possible that these p16+ cells are cleared but rapidly repopulate, as the mice are harvested 4-6 days following the last AP dose. This was also apparent in the CD24[low] osteolineage population, which exhibited higher overall

Ki67+ cells than the CD24[high] osteolineage population (Supplementary Fig. 8C). Further studies are needed, however, to address this issue. In addition to clearance of these clusters, the early osteoblast cluster increased in number after AP treatment (Fig. 3F, G), suggesting that this is a newly replenished osteogenic population that appears after the clearance of senescent cells. This is supported by pseudotime analyses, which indicated that there is an emergence of cells that are early in the differentiation continuum following clearance of p16+ senescent cells (Supplementary Fig. 8D). This finding is entirely consistent with our previous work which demonstrated that AP treatment in aged *INK-ATTAC* mice improved endosteal osteoblast numbers and bone formation rates[16]. Finally, to support these findings we performed CITRUS analysis, which generates separately stratified clusters from the original dataset to observe statistical differences, and independently found downregulation of CD24+ clusters with AP treatment, along with an increase in Runx2+ and Runx2/Osterix+ early osteoblast clusters (Supplementary Fig. 8E).

Importantly, as noted above, we have previously demonstrated senescence within osteocytes with age – and their clearance with senolytic therapy – through gene expression and histological senescent phenotyping[16]. Entirely consistent with this, we now demonstrate concomitant increases in p16 and BCL-2 expression (Fig. 3C) and in % p16KB cells (Fig. 3D) in the late osteoblast/osteocyte cells, along with clearance of these cells following AP20187 treatment (Fig. 3E–G). In addition, we identify populations (CD24[high] osteolineage and CD24+Osterix+ cells) that also demonstrate increases in %p16KB cells with age (Fig. 3C, D) and are targets of senolytic clearance (Fig. 3E–G). Interestingly, in aged mice, CD24[high] osteolineage cells displayed the highest expression levels of markers for senescence, SASP, and DNA damage, with the CD24+Osterix+ cells being the lowest, and the late osteoblast/osteolineage cells having intermediate levels of these markers (Fig. 3J–L). These data thus reveal that CD24[high] osteolineage cells exhibit a robust senescent and inflammatory profile relative to other skeletal populations and · in addition to osteocytes, as previously reported[16] - these CD24[high] osteolineage cells are key targets of p16-driven senolytic clearance.

## CD24 osteolineage populations exhibit a unique SASP profile with age
To further investigate the senescent profile of CD24+ osteolineage cells, we applied CITRUS analysis and compared this to our established FlowSOM clusters (Fig. 4A). Consistent with the findings noted above, we found that many clusters exhibited higher expression levels of senescence/SASP proteins in old compared to young mice (FDR < 5%) (Supplementary Fig. 8F). Of these proteins, p16 and BCL-2 were upregulated in the largest number of clusters with age, supporting their use as robust predictors of senescence (Fig. 4B). These factors were followed closely by CXCL1, FLAG (detecting the p16[Ink4a] promoter activity in the *INK-ATTAC* transgene) and several interleukins (IL-6, IL-1α, IL1β). As observed previously, p21 was not found to be differentially regulated with age by CITRUS.

Among the clusters with increased p16 expression with age, cells expressing CD24/Runx2/Osterix exhibited a unique upregulation of

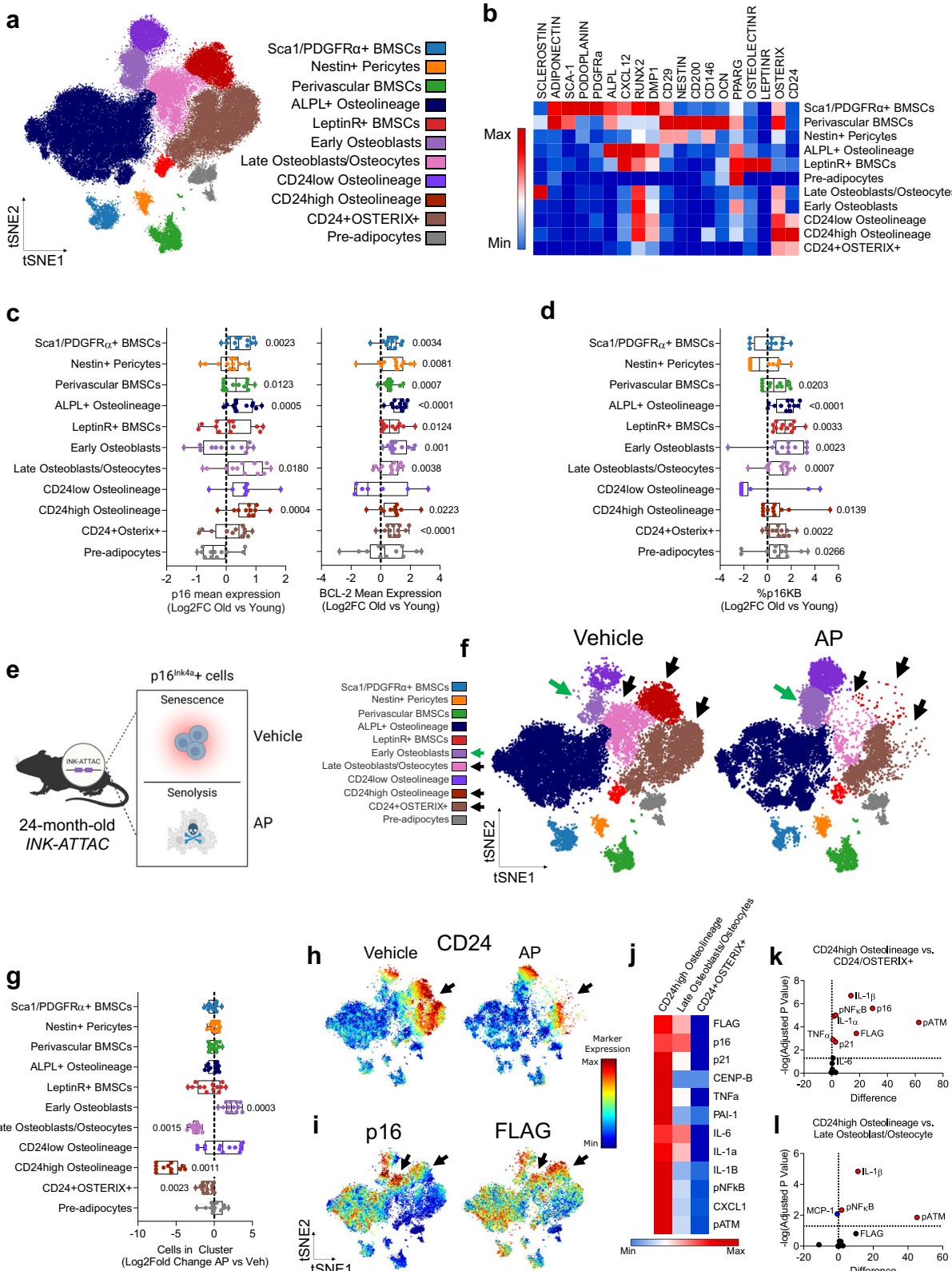

the most SASP inflammatory proteins, including IL-1α, IL1β, CXCL1, and TNFα (Fig. 4C, D); these cells also exhibited robust upregulation of BCL-2 and an increased percentage of p16KB cells with age (Fig. 4D). When these CD24/Runx2/Osterix cells were overlaid with our original t-SNE plots, they were largely contained within CD24+ osteolineage clusters, along with partial overlay in the late osteoblast/osteocyte cluster (Fig. 4E). This independent CITRUS analysis thus strongly

supports our previous data demonstrating increased senescent cell burden in these clusters with age (Fig. 3C, D) and their clearance with AP (Fig. 3F).

We next investigated our scRNA-seq dataset, finding that CD24 osteolineage cells (positive for *Cd24a* [CD24] and *Runx2*) strongly co-expressed not only *Bcl2*, but also other anti-apoptotic genes *Bcl2l1* (BCL-XL) and *Bcl2l2* (BCL-W) (Fig. 4F, G). This suggests that

**Fig. 3 | CD24$^{high}$ osteolineage cells represent inflammatory senescent cells in old mice targeted by genetic senolytic clearance. a** t-SNE visualization and FlowSOM clustering of $n = 80,000$ CD45-Lin- bone and marrow cells ($n = 40$ INK-ATTAC mice [$n = 15$ young, $n = 12$ old + vehicle, $n = 13$ old + AP] – 2,000 cells sampled per mouse) analyzed by CyTOF. Cells are colored by clustered population (see Table 1 for defining markers); **b** Heatmap representation of the 11 cell clusters and protein expression of identification markers; **c** Log2 fold-Change mean expression of p16 or BCL-2 in each cluster with age. **d** Log2 fold-change of %p16KB cells in each cluster across aging; **e** Schematic of p16+ senescent cell clearance in 24-month-old AP-treated *INK-ATTAC* mice; **f** t-SNE plots of FlowSOM clusters of bone/bone marrow cells from old vehicle- or AP-treated *INK-ATTAC* mice. Black arrows indicate cleared clusters while green arrows indicate increased cluster abundance; **g** Quantification of cluster abundance changes between vehicle- and AP-treated mice (Log2-fold change to vehicle); **h** CD24 expression feature plots in samples from vehicle- or AP-treated mice; **i** Feature plots of p16 and FLAG (*INK-ATTAC* transgene) protein expression; **j** Heatmap and **k**, **l** volcano plots for statistical comparisons of senescence marker expression between cleared clusters in vehicle-treated old mice. Schematic in (**e**) was generated using BioRender. Box plots show median and interquartile ranges with error bars representing minimum and maximum values. (**c**, **d**, **g**) Two-sided Mann–Whitney test or unpaired t test, as appropriate. **k**, **l** Multiple two-sided t tests with Holm-Sidak Correction. Source data are provided as a Source Data file.

in addition to their inflammatory profile, CD24 osteolineage cells are strongly resistant to apoptosis, perhaps through several mechanisms. Collectively, we further establish that CD24 defines a senescent osteolineage population exhibiting an expression profile enriched for SASP and apoptosis-resistance factors in the aging bone microenvironment.

### Pharmacological senolytic treatment also targets CD24$^{high}$ osteolineage cells

The unique inflammatory profile of senescent CD24+ osteolineage cells led us to test if they are targeted not only by genetic but also by pharmacological senolytic treatment, which selectively kills cells based on senescent cell anti-apoptotic pathways (SCAPs) rather than by activation of transgenic caspase 8 in p16+ cells (as done in *INK-ATTAC*

**Table 2 | Defining markers for CyTOF cluster identification**

| Cluster name | Defining markers | References | Additional markers |
|---|---|---|---|
| Sca-1/PDGFRα + BMSCs | Sca-1 PDGFRα | 58 59 | PDPN[60,67] CXCL12[123] Adiponectin[66,124] Runx2[125] CD29[58] Dmp1[126] ALPL[127] |
| Nestin+ Pericytes | Nestin CD146 | 63 | CD29[128] |
| Perivascular BMSCs | CD146 Sca-1 CD200 CD29 | 61,62 | Adiponectin[66,124] OCN[126] Osterix[129,130] |
| ALPL+ Osteolineage | ALPL Runx2 | 70,71 | Dmp1 CXCL12[37,123] |
| LeptinR+ BMSCs | LeptinR | 57,61 | OsteolectinR[65] CXCL12[57,131] Runx2[132] Pparg[133] |
| Early Osteoblasts | Runx2 Osterix | 68,69 | N/A |
| Late Osteoblasts / Osteocytes | Sclerostin Osterix Runx2 | 72,81 | Dmp1[134] |
| CD24low Osteolineage | CD24(low) Osterix Runx2 | 75–77 | N/A |
| CD24high Osteolineage | CD24(high) Osterix Runx2 | | |
| CD24+/OSTERIX+ | CD24 Osterix | | N/A |
| Pre-adipocytes | Pparg | 73 | N/A |

Each cluster was defined by expression of specific defining markers and additional markers as characterized by previous studies in the literature.

mice). Thus, we performed CyTOF on aged *C57BL/6N* wild-type mice treated with or without Dasatinib + Quercetin (D + Q) (Fig. 5A): a combination senolytic therapy that targets SCAPs and which we have previously shown to reduce frailty and prevent bone loss in aged mice[14,16,85]. Multidimensional analysis from this cohort identified skeletal cell populations consistent with our *INK-ATTAC* cohort (Supplementary Fig. 9A, B), and treatment with D + Q similarly targeted CD24$^{high}$ osteolineage, CD24+/Runx2+, and late osteoblast/osteocyte clusters, although the reduction in the latter cells was modest (Fig. 5B). Independently, with CITRUS analysis we found that D + Q was more limited than the genetic *INK-ATTAC* model in its clearance, only targeting 5 clusters overall (Fig. 5C). These clusters were all high in expression for CD24 and represented two separate families of CD24+/Osterix+/Runx2+ and CD24+/Osterix+ clusters. In sequential analyses, we found that D + Q reduced expression of p16, SASP factors PAI-1 and IL-1β, and the DNA damage marker pATM within CD24/Runx2/Osterix+ cells (Fig. 5D; Supplementary Fig. 9C). This aligns with the established effectiveness of D + Q on senescent cells exhibiting DNA damage and serpine (PAI) family proteins[85].

Due to the recurring presence of CD24 on senescent cell clusters, we next sought to determine if this marker can be used to enrich for cells susceptible to senolytic clearance. Using manual gating on our CyTOF data, we found that total CD45-CD24+ stromal cells were reduced in both genetic (*INK-ATTAC*) and more modestly by pharmacological (D + Q) methods of senolytic clearance, while CD24- cells were unaffected (Fig. 5E, F). Importantly, CD24+ cells that were CD45+ were not cleared (Fig. 5G), demonstrating senolytic specificity for non-hematopoietic, mesenchymal CD24+ cells. Overall, we establish that CD24 is expressed by aged skeletal senescent cells that are cleared by both genetic and pharmacologic senolytic therapy.

### CD24+ cells display functional characteristics of senescence

To further test if CD24+ skeletal cells are enriched for growth-arrested senescent cells, we isolated non-hematopoietic (CD45-/Lin-) CD24+ and CD24- cells from the digested bone and marrow of aged mice for in vitro phenotyping (Fig. 6A; See Supplementary Fig. 9D for gating strategy). Cell cycle analyses revealed that CD24+ cells were largely growth-arrested, with a larger proportion of cells in the G0/1 phase and less in the G2/M phase than CD24- cells (Fig. 6B, C). When placed in culture, CD24+ cells exhibited markedly reduced colony-forming efficiency (CFE) after 7 days, while CD24- cells grew rapidly (Fig. 6D, E). CD24+ cells generated very small colonies, typically with only a few cells, demonstrating impaired stemness and proliferative capabilities. Moreover, CD24+ cells exhibited spontaneous senescence, with up to 40% of cells staining positive for senescence-associated β-galactosidase (SA-β-gal) after only 14 days in culture, while CD24- cells continued to proliferate (Fig. 6E).

Although co-expressed with osteogenic markers in both scRNA-seq and CyTOF data, it remains unclear if CD24+ cells have osteogenic capabilities. Thus, we induced osteogenesis in both CD24+ and CD24- cells (Fig. 6F), finding that CD24+ cells have limited osteogenic potential (Fig. 6G). CD24- cells underwent robust osteogenesis,

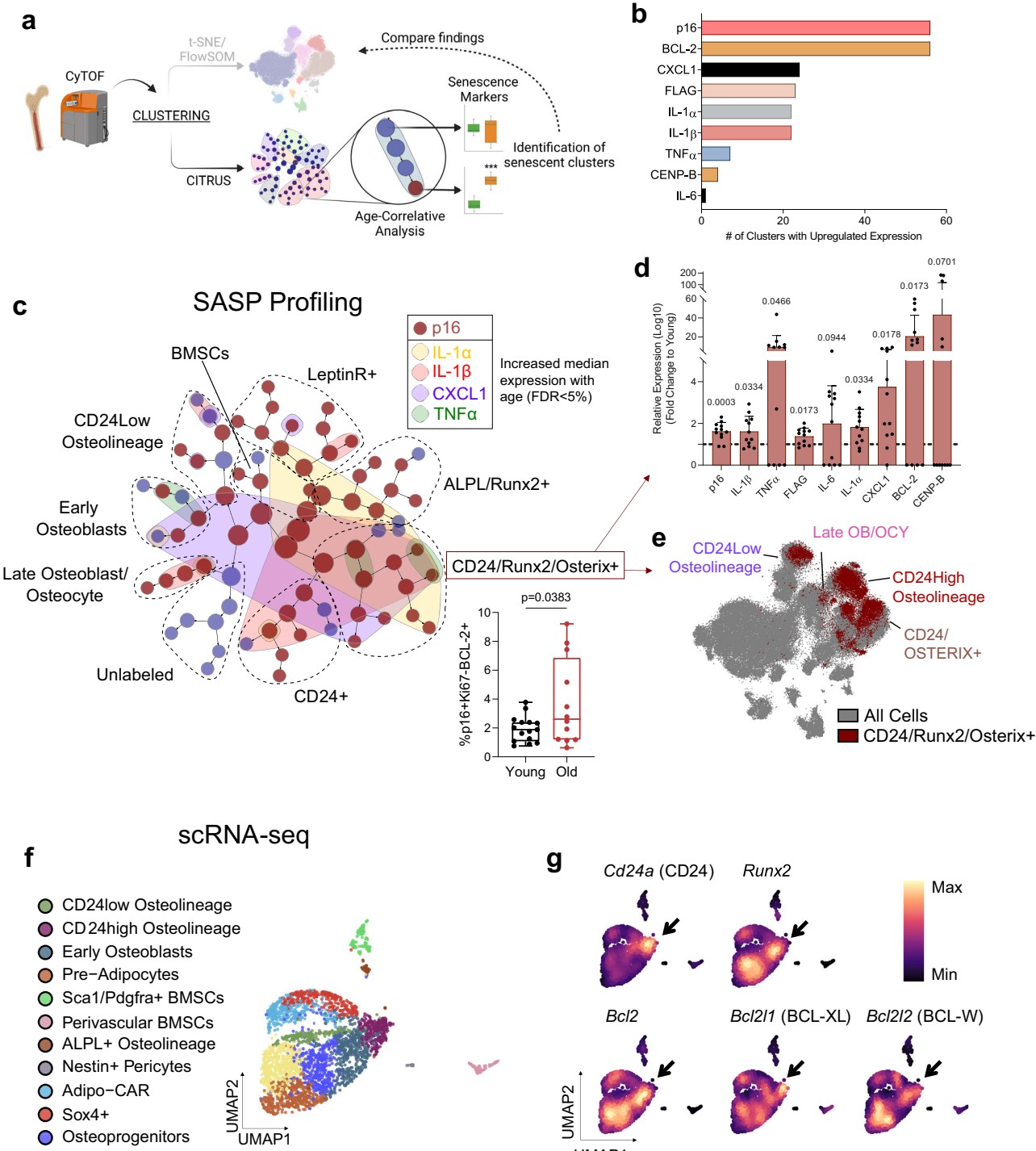

**Fig. 4 | CD24 osteolineage cells display an age-related senescence expression profile. a** Schematic of CITRUS analysis workflow; **b** Aging markers upregulated in the highest number of CITRUS clusters with age (FDR < 5%); **c** Senescence/SASP marker expression changes across aging overlaid on CITRUS plot (each circle is a cluster, grouped by cluster families dotted lines; see Supplementary Fig. 6A for identity marker expression). Red clusters indicate upregulated p16 expression, while colored overlays indicate upregulation in the corresponding SASP marker. All markers converge in CD24+/Runx2+/Osterix+ clusters, indicating that this population exhibits the most widespread age-related increase in SASP marker expression in bone/bone marrow. Box plot below demonstrates the upregulation of %p16KB cells with age in this population. Red arrows point to **d** quantified median expression of various SASP factors in the CD24+/Runx2+/Osterix+ clusters with age (mean ± SD) and **e** overlay of the CD24+/Runx2+/Osterix+ cluster on original t-SNE plot with FlowSOM clusters labeled (**c, d**) n = 15 young and n = 12 old biologically independent animals; **f** UMAP visualization by scRNA-seq of n = 3,362 clustered Lin-CD45- cells from the digested bone and marrow of n = 3 24-month untreated male *INK-ATTAC* mice; CAR, Cxcl12-abundant reticular; **g** Density plots of CD24 osteolineage markers and anti-apoptotic factors (*Bcl2, Bcl2l1, Bcl2l2*). Schematic in (**a**) was generated using BioRender. Box plots show median and interquartile range with error bars representing minimum and maximum values. **c** Two-sided Mann–Whitney test. **d** Multiple two-sided t-tests with Holm-Sidak Correction. Source data are provided as a Source Data file.

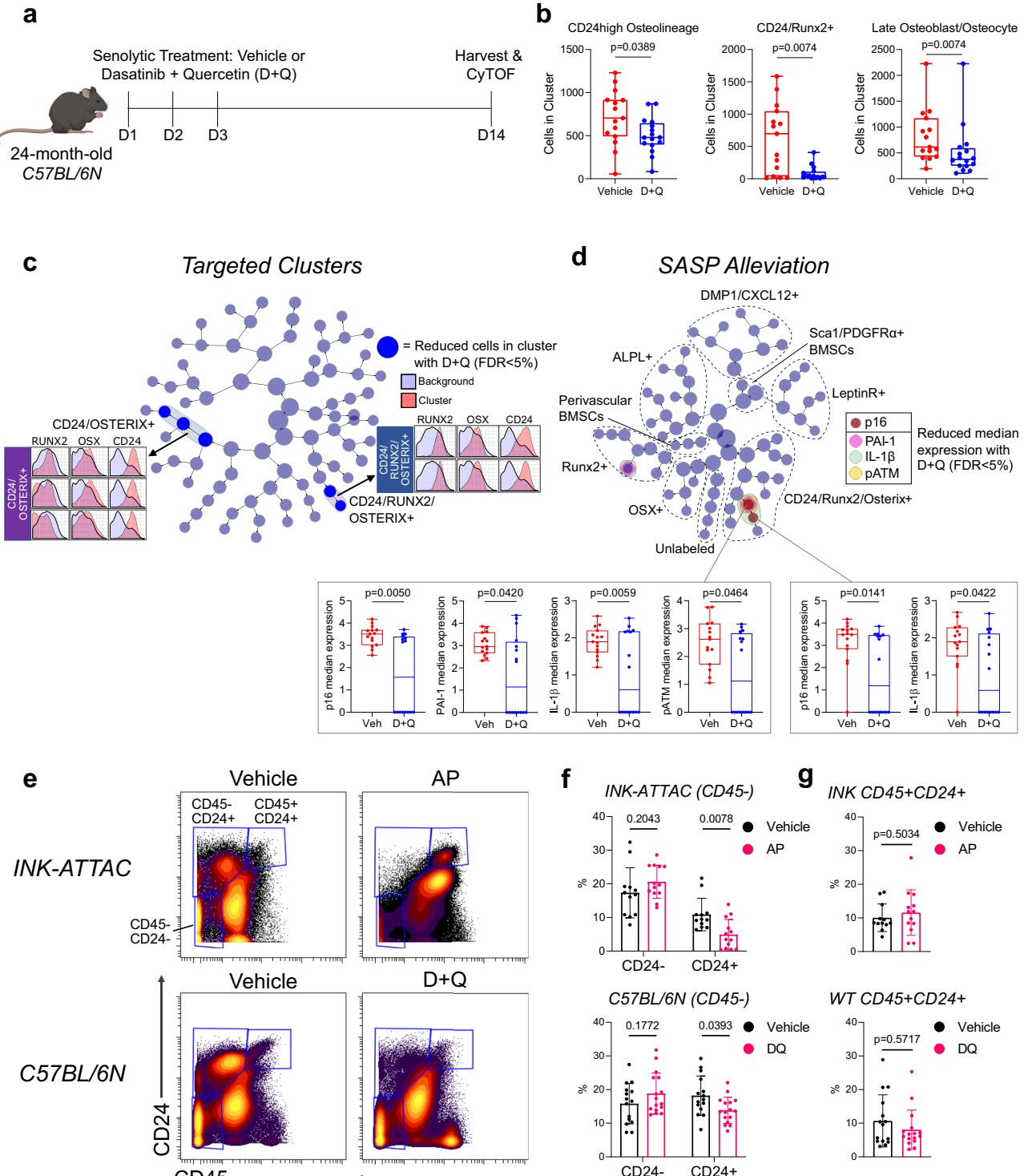

**Fig. 5 | Pharmacologic senolytic therapy targets CD24+ osteolineage cells in aged mice. a** Experimental design of pharmacological senolytic treatment of 24-month-old C57BL/6 N mice with Dasatinib + Quercetin (D + Q). Mice were treated for 3 consecutive days (*n* = 15 old + vehicle and *n* = 16 old + D + Q biologically independent animals), then harvested at 14 days (indicated by dashes); **b** Quantified cell abundance changes with D + Q treatment in clusters similarly targeted in *INK-ATTAC* mice (See Supplementary Fig. 9A, B for cluster definitions); **c, d** CITRUS analysis of cluster abundance (**c**) and median expression changes (**d**) between vehicle- and D + Q-treated mice (FDR < 5%). **c** Cleared clusters, marked by blue, are defined by high CD24, Osterix, and/or Runx2 expression, as shown by histograms. **d** Clusters with reduced median expression are colored by their respective marker, with box plots demonstrating expression changes (Mann–Whitney test). Cluster

families are marked by dotted lines (See Supplementary Fig. 9C for defining markers); **e** Gating strategy for CD45-CD24+ and – populations from all cells in CyTOF data from *INK-ATTAC* mice treated with vehicle or AP (*n* = 12 vehicle, *n* = 13 AP), and C57BL/6 N mice treated with vehicle or D + Q (*n* = 15 vehicle and *n* = 16 D + Q); **f** Quantification of cell population percentages demonstrate both senolytic treatments clear CD45-CD24+ cells, but not CD45-CD24- or (**g**) CD45 + CD24+ cells (mean ± SD). Schematic in (**a**) was generated using BioRender. Box plots show median and interquartile range with error bars representing minimum and maximum values. **b, g** Two-sided Mann–Whitney or unpaired t test, as appropriate; **f** Multiple two-sided t tests with Holm–Sidak Correction. Source data are provided as a Source Data file.

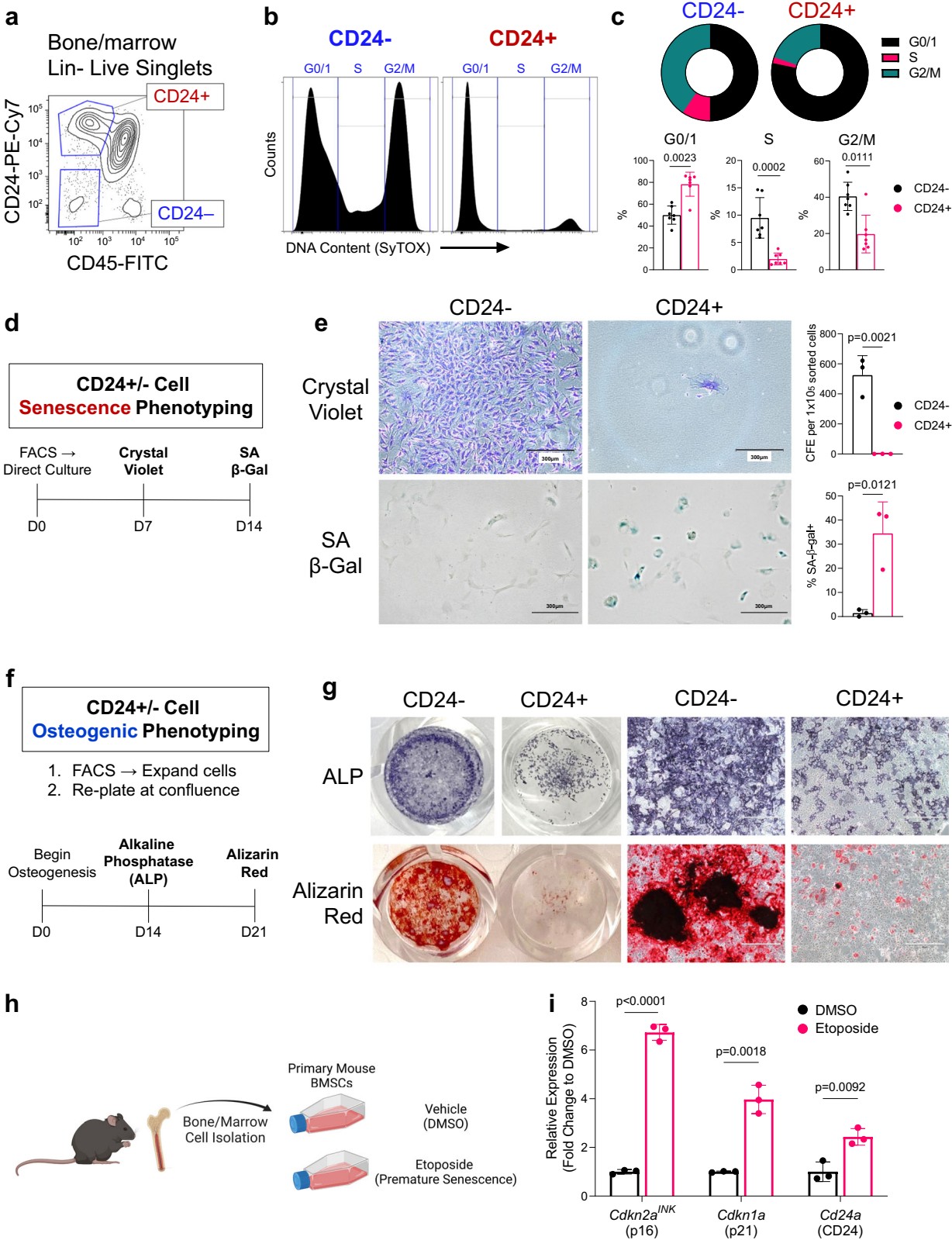

staining strongly for alkaline phosphatase (ALP), a measure of osteoblast differentiation, and Alizarin Red, which stains mineralized nodules. In comparison, CD24+ cells exhibited impaired ALP and Alizarin Red staining, even with sufficient cell numbers (Fig. 6G). Interestingly, these in vitro findings are consistent with the in vivo pseudotime analysis in Supplementary Fig. 6A which shows that the CD24+ osteolineage cells formed their own bifurcation distinct from

the typical differentiation of BMSCs to late osteoblasts and osteocytes. We acknowledge, however, that the impaired differentiation of the CD24+ cells could also be due to their post-mitotic status, although these cells not only exhibited impaired osteogenic capabilities but were highly positive for the senescence marker, SA-β-gal.

In various cancers, CD24 serves as a stress marker indicative of tumor burden, rather than a cell-specific marker[86–88], which led us to

**Fig. 6 | Isolated CD24+ bone stromal cells exhibit functional characteristics of senescence and impaired osteogenesis. a** Gating strategy for FACS isolation of CD24+ and CD24− skeletal stromal cells from Lin-depleted bone/marrow cell suspensions; **b** Gating strategy and **c** quantification of cell cycle analysis of CD24+/− cells (*n* = 7 mice); **d** Outline of in vitro senescence phenotyping of CD24+/− cells; I Brightfield images and quantification of colony formation efficiency (CFE) assay and SA-β-gal staining of CD24- and CD24+ cells after 7 or 14 days in culture, respectively (*n* = 3 mice); **f** Outline of osteoblast differentiation assays of CD24+/− cells; **g** CD24- and + cell monolayers stained with either alkaline phosphatase (ALP)

or Alizarin Red after 14 and 21 days in culture, respectively. Accompanying high magnification images are 20X (*n* = 3 mice); **h** Schematic of in vitro etoposide-induced senescence of BMSCs; **I** MRNA levels of *Cdkn2a^{INK}* (p16), *Cdkn1a* (p21), and *Cd24a* (CD24) from BMSCs treated with vehicle (DMSO) or etoposide (*n* = 3 mice). Schematic in (**h**) was generated using BioRender. Scale bars represent 750 μm. Bars show mean ± SD (**c**, **e**). Two-sided unpaired t-test or Mann−Whitney test as appropriate; (**i**) Multiple two-sided t tests with Holm−Sidak Correction. Source data are provided as a Source Data file.

test if CD24 may serve a similar role in senescence. We found that CD24 is induced in the senescence program, as in vitro etoposide-induced senescence of BMSCs led to elevated *Cd24a* expression, alongside *Cdkn2a* (p16) and *Cdkn1a* (p21) (Fig. 6H−I). Taken together, these data demonstrate that CD24+ stromal cells exhibit functional characteristics of senescence and impaired osteogenesis, and CD24 may label cells prone to senescence and senolytic clearance due to its upregulation during the onset of senescence.

## Discussion

As senescent cells cannot be defined by a singular marker[89], the identification of these cells in vivo has been impeded by technical limitations. Here, we provide an approach to identify and characterize in vivo senescent mesenchymal cells using multiplexed cellular profiling by CyTOF. Using a methodically validated antibody panel, we dissected senescent cell heterogeneity and defined specific cell populations fulfilling major requirements for senescence. We then applied this definition to aging skeletal cell populations to provide insights into the identity and characteristics of senescent mesenchymal cells in the bone microenvironment, uncovering physiologically relevant cell types that are targeted by senolytic treatments we previously demonstrated to be effective in preventing age-related bone loss in mice[16].

Although long-established as a marker of senescence, we found that p16 expression labels a diverse set of cells with both senescent and non-senescent properties. This finding is entirely consistent with recent findings using highly sensitive p16^{Ink4a} reporter mice (INKBRITE)[39], although we provide important complementary data at the protein level to the transcriptional data in the previous study. Specifically, similar to our finding of p16 protein expression in both Ki67 negative and positive bone and marrow mesenchymal cells, in the INKBRITE p16^{Ink4a} reporter mice, p16^{Ink4a} was also expressed in both proliferating and non-proliferating lung fibroblasts. Importantly, and entirely consistent with our findings showing an inverse association between p16 and Ki67 protein expression (Fig. 2D−F; Supplementary Fig. 3B), high p16^{Ink4a}-expressing fibroblasts from the INKBRITE mice had lower proliferative capacity than low p16^{Ink4a}-expressing cells[39]. Collectively, these findings indicate that there is a spectrum of p16^{Ink4a} RNA and protein expression in vivo, with high expression associated with growth arrest and the full senescent phenotype.

Our finding that BCL-2 co-expression is required to define senescent p16+ cells that are also growth-arrested (Ki67-) aided in our further defining of senescent skeletal cell types, without additional noise through comparing total p16 positivity. Moreover, our strict validation of the p16 antibody for antigen specificity, sensitivity, and biological signal permitted the in-depth characterization of these cells. The application of this CyTOF panel therefore could advance the study of senescence in various murine disease states without the need for genetic reporters.

Like p16+ cells, p21+ cell subsets positive for BCL-2 were associated with age in skeletal mesenchymal cells, even though total p21+ cells were not. Although this was unexpected, as p21 has long been established as a marker for senescence[29,90–92], this further exemplifies the ability of BCL-2 co-expression to define senescent cells. Furthermore, although this work focused on p16KB cells due to their

higher abundance, greater association with age, and previous studies demonstrating beneficial skeletal effects in aged mice of clearing p16+ cells[16] it will be of interest for future studies to define the function of p21KB cells in skeletal aging. Recent work has demonstrated that p21+ senescent cells are causal to radiation-induced bone loss, while p16+ senescent cells are dispensable[93]. These data suggest the hypothesis that, in bone (and perhaps in other tissues), p16KB cells contribute to age-related senescence, while p21KB cells, albeit still senescent, may have more of a prominent role in acute senescence caused by injury. This is supported by recent work demonstrating that p21 expression is upregulated in bone fracture, with its highest expression immediately following injury, and then waning as p16 upregulation emerges[94].

Along with verification of senescence in osteocytes, as shown previously, our work also establishes CD24^{high} osteolineage cells as a previously uncharacterized senescent skeletal cell population in aged mice, which we characterized both in vivo and in vitro. Isolated CD24+ cells exhibited an enrichment of senescent cells and had limited osteogenic capabilities. In vivo, we found that the CD24^{high} osteolineage cells had a marked upregulation of SASP factors, and clearance of these cells resulted in an increase in early osteoblast lineage cells, consistent with our previously observed effects of senescent cell clearance on improving bone formation in aged mice[16]. Notably, as revealed by our CITE-Seq analyses, the expression of early (*Runx2*), mid (Sp7), and late (*Bglap*) osteogenic genes within CD24+ cells clouds whether these cells are progenitors or late osteogenic cells, or perhaps a heterogeneous mix of cells at various stages of osteoblastic differentiation. Along these lines, we cannot exclude the possibility that genetic senolysis preferentially removes cells of the osteoblast lineage at the terminal stage of differentiation, regardless of CD24 status, resulting in the elimination of late osteoblastic cells expressing CD24 (although osteocytes do not appear to express CD24, Supplementary Fig. 5C). Thus, additional in vitro and in vivo studies are needed to further define the regulation of CD24 expression in senescent osteolineage cells as well as possible paracrine effects of these CD24 osteolineage cells on osteoblasts as well as osteoclasts and other cells in the bone microenvironment, including adipocytes and hematopoietic cells.

CD24 is a cell-surface marker with established roles in stress and immune signaling[86,87,95–98], yet ours is the first work, to our knowledge, to implicate CD24 in senescence in vivo. In cancer, CD24 expression is strongly associated with reduced life expectancy and is used as a diagnostic marker for patient prognosis[88]. Therefore, based on the evidence presented here, it is plausible that this role of CD24 is conserved in senescent cells, and CD24 expression may have potential as a diagnostic marker for senescence. It is important to note, however, that as the senolytic effect was exclusive to non-hematopoietic cells (Fig. 5E−G) and CD24 is expressed on many immune cell types (particularly B-cells)[79,98], its application to detect senescent cells will likely only apply to mesenchymal cells. The advantage compared to current senescence biomarkers (p16, p21) is that CD24 is a cell surface protein, which permits simultaneous mesenchymal cell purification alongside tracking, targeting, and sorting live senescent cells through cytometric methods. Thus, the implementation of this marker may have important implications for pre-clinical senolytic screens in murine models,

and perhaps even validation of senolysis in human clinical studies. We acknowledge, however, that additional studies are needed to evaluate the potential role of CD24 in marking senescent cells in other tissues beyond the bone microenvironment and across species. Moreover, senolytics potentially targeting CD24-expressing cells would require careful evaluation as non-senescent CD24+ cells may also have beneficial effects across tissues.

An important feature of our study is that it builds upon transcriptomic studies through investigating protein expression, which has several advantages: protein is more stable and has a longer half-life than RNA[99–102], protein expression is more conserved than mRNA among species[103], and mRNA transcription does not necessarily predict protein translation[101,104]. Therefore, the application of CyTOF, perhaps even in combination with scRNA-seq, allows for the rigorous investigation of senescent cells at the single-cell resolution. Importantly, our validation of CyTOF antibodies, particularly p16, and p21, provides confidence that this panel can reproducibly detect senescent cells in various applications. We expect that this established workflow can be utilized to delineate tissue-specific senescent cell identities and overcome the technical challenges of senescent cell handling and antibody specificity.

We acknowledge the potential limitations of our study. Specifically, CyTOF relies on a pre-specified panel of antibodies, which limits the exploration of further populations. This prevented the inclusion of additional markers, e.g. for cell identity[105–107] and senescence[89,108,109]. We also acknowledge that there may well be further heterogeneity within our working classification of bone/marrow mesenchymal cell populations. Specifically, the factors defining CD24+ osteolineage subpopulations (e.g., CD24high/low osteolineage and CD24+Osterix+) remain unclear, and it will be important to further characterize markers that can identify CD24+ subsets that are highly susceptible to senescence and senolytic clearance. However, in addition to being consistent with the existing literature on the identity of these cell populations, this classification does provide a useful framework for the subsequent analyses focusing on senescent cells by at least providing a provisional identity to these cells. We sought to address this limitation with the inclusion of scRNA-seq analysis, and we observed similarly clustered populations with both analyses. Nonetheless, it will be of interest to define newly discovered senescence markers and cell types, particularly those difficult to isolate in suspension (e.g., deeply embedded osteocytes), as our understanding of the aging bone microenvironment advances further. Finally, it is important to acknowledge that our characterization of p16KB and p21KB senescent cells applies only to mesenchymal cells; whether "senescent" immune cells exhibit similar characteristics or not remains to be defined.

In summary, we provide an approach to define senescent cells in vivo at the single-cell level using multiplexed protein profiling by CyTOF. Importantly, our definition of senescent cells is entirely consistent with a recent consensus from the ICSA[56] and includes p16 or p21 positivity, growth arrest, upregulation of anti-apoptosis pathways, expression of a SASP, evidence of DNA damage, and virtual absence in youth with a marked increase with aging – all of which we defined within the same cells. From a translational perspective, we provide a deeper characterization of specific cell types targeted by senolytic clearance, including identifying a specific CD24high osteolineage population, thereby facilitating the ultimate goal of defining better targets and approaches for the treatment of osteoporosis. In addition, the unique property of BCL-2 to identify true aging-specific senescent cells strongly support the development of new BCL-2 (and perhaps other BCL-related protein) inhibitors with optimized side-effect profiles to specifically target aged senescent cells for clearance. Finally, our work also points to the potential utility of CD24 for the identification and tracking of senescent cells in the bone microenvironment and perhaps in other tissues as a tool to evaluate the efficacy of senolytic treatments.

## Methods

### Animals
All animal studies were performed under protocols approved by the Institutional Animal Care and Use Committee (IACUC), and experiments were performed in accordance with Mayo Clinic IACUC guidelines. Mice were housed in ventilated cages and maintained within a pathogen-free, accredited facility with constant temperature (25 °C), 30–70% humidity, a twelve-hour light/dark cycle, and access to food and water *ad libitum*. Mice used included *INK-ATTAC*[12], *C57BL/6N* Wild Type (WT), p16 luciferase reporter mice (*p16Luc*)[38], Runx2-Cre[110], and TdTomato (B6;129S6-Gt(ROSA)26Sor^tm9(CAG-tdTomato)Hze/JAi9; JAX #7909) mouse models. Two ages were used for all studies: Young-6 month, and old-24 month. All young mice were untreated. Old *INK-ATTAC* mice were randomized by weight for treatment with either vehicle (4%ETOH 10%PEG-400 and 2%Tween) or 10 mg/kg AP dissolved in vehicle, administered subcutaneously twice weekly for two weeks. Old C57BL/6 N mice were randomized by weight for treatment with vehicle (10% EtOH, 30% PEG-400, 60% Phosal-50) or Dasatinib + Quercetin (D + Q) (Dasatinib 5 mg/kg and Quercetin 50 mg/kg) dissolved in vehicle, administered by oral gavage for three consecutive days and harvested at two weeks post-treatment. Each cohort consisted of the following groups: *INK-ATTAC*: Young (n = 15; 10 female, 5 male), Old + Vehicle (n = 12; 6 female, 6 male), Old + AP (n = 13; 6 female, 7 male). C57BL/6 N: Young (n = 17; 10 female, 7 male), Old + Vehicle (n = 15; 8 female, 7 male), Old + D + Q (n = 16; 10 female, 8 male). In comparisons of young versus old mice, Old + Vehicle mice were used, for reduction purposes. To our knowledge, there are no effects of vehicle administration for either treatment on aging, senescent, or skeletal outcomes. For CD24 cell isolation and scRNA-seq studies, 24-27-month untreated male and female INK-ATTAC mice were used, with sexes indicated in figure legends per experiments.

### Consideration of sex as a biological variable
Per NIH guidelines[111,112], we studied both female and male mice. In order to test for possible effects of sex on our primary endpoints[113], we performed 2-way ANOVA tests on several important parameters of aging and senolytic treatment (Supplementary Table 1). We found that neither sex alone nor the interaction between sex and age was significant, indicating that these cellular effects of aging or senolytic treatment are not dependent on sex. Thus, both males and females were analyzed together.

### Dissociation and purification of mesenchymal cells from skeletal tissue
Mice were euthanized according to standardized and approved IACUC protocols. Femurs and tibiae were isolated, cleaned of soft tissue, cut at both ends, and marrow centrifuged out of the diaphyses and metaphyses into a collection tube. Marrow was resuspended in 1 mg/mL Liberase DL (Sigma) diluted in FACS buffer (0.5% BSA [Sigma] in PBS) and digested at 37 °C for 30 min to increase yield of stromal cells released from the vasculature fraction as previously described[33]. Diaphyses and metaphyses cleared of bone marrow were gently crushed, rinsed in PBS, and then digested in 300 Units/mL of Collagenase IA (Sigma), diluted in MEM α (ThermoFisher), 3 times for 25 min each. Bone and marrow solutions were then combined and treated with RBC lysis buffer (ThermoFisher) to clear erythrocytes. The sample was then depleted of cells expressing hematopoietic lineage markers (CD5, CD45R [B220], CD11b, Gr-1 [Ly-6G/C], 7−4, and Ter-119) using Magnet Assisted Cell Sorting (MACS) and the Lineage Cell Depletion Kit (Miltenyi Biotec).

### Cell culture
Primary mouse BMSCs were generated from 6-month-old C57BL/6 N mice by digesting freshly dissected femurs and tibias 3 times for 25 min each, followed by RBC lysis, as described above. After expansion in

growth media (DMEM [ThermoFisher] + 15% FBS [Gemini Bio-Products] + 1X Antibiotic/Antimycotic [ThermoFisher] + 1X Gentamicin [Sigma]) in hypoxic condition (2% O2), BMSCs were seeded at $4 \times 10^4$ cells/cm$^2$ in 75cm$^2$ flasks and treated for 48 h with either vehicle (0.1% DMSO [Sigma]) or 20uM of etoposide (MilliporeSigma, St. Lous, MO) dissolved in vehicle, followed by maintenance in growth media for 6 days. Cells were then dissociated using Trypsin-EDTA (Gibco) and processed for CyTOF. CD24- and + cells were maintained in the same growth media, as described above. For osteogenesis experiments, CD24- and + cells were seeded at $2 \times 10^4$ cells/cm$^2$ in 96-well plates, upon which media was changed to osteogenic medium (MEM α + 10% FBS + 1X Anti/Anti + 10 mM β-Glycerophosphate [Sigma] + 50 mg/ml Ascorbic Acid [Sigma]) and differentiated until the indicated time point, changing media every 48 h. U2OS cells were maintained in normoxic conditions (5% O2) in 75 cm$^2$ flasks in DMEM + 10% FBS + 1X Antibiotic/Antimycotic and seeded at $2 \times 10^4$ cells/cm$^2$ in 6-well plates for transfection. Expression constructs were transfected into U2OS cells using 1.5 μg of DNA and 4.5 μL FuGENE 6 Transfection Reagent (Promega) per well. DNA-FuGENE mixtures were combined in Opti-MEM (ThermoFisher), then added dropwise to U2OS cells. After 24 h, cells were lifted using Trypsin-EDTA and processed for CyTOF, while separately lysing 10% of each cell sample in QIAzol for qPCR validation.

## p16 expression vector construction

The pcDNA5/TO-p16ink4a plasmid was constructed by cloning the open reading frame of mouse p16$^{Ink4a}$ into the BamHI site of pcDNA5/TO (Promega, Madison, WI).

## CyTOF processing

Custom conjugated antibodies were generated in-house through the Mayo Clinic Hybridoma Core using Maxpar X8 Ab labeling kits (Standard BioTools) according to the manufacturer's protocol. Conjugated antibody concentrations were measured by absorbance at OD280nm and then normalized to a 5 μg/μL stock concentration. Isolated cells were resuspended in 1 mL of Cell Staining Buffer (CSB) (Standard BioTools) and incubated for 5 min with 0.5 μm Cisplatin solution (Standard BioTools) in PBS. Samples were then washed twice with CSB. An antibody cocktail of the entire phenotyping panel (individual antibodies used, including the p16 antibody [EPR20418, ab232402, Abcam], and their dilutions are listed in Supplementary Table 3) was prepared as a master mix prior to adding 50 μL of cocktail to samples resuspended in 50 μL of CSB. Samples were then incubated at room temperature for 45 min. Samples were washed twice and then fixed with 2% PFA (Standard BioTools) in PBS. After fixation and wash, samples were resuspended in 30 nM intercalation solution (Standard BioTools) and incubated overnight at 4 °C. On the following morning, cells were washed with PBS and resuspended in a 1:10 solution of calibration beads and cell acquisition solution (CAS) (Standard BioTools) at a concentration of $0.5 \times 10^6$ cells/mL. Prior to data acquisition, samples were filtered through a 35 μm blue cap tube (Falcon). The sample was loaded onto a Helios CyTOF system (Standard BioTools) and acquired at a rate of 200–400 events per second. Data were collected as.FCS files using the Cytof software (Version 6.7.1014). After acquisition, intra-file signal drift was normalized to acquired calibration bead signal using Cytof software.

## CyTOF data analysis

**Initial processing and clustering.** Cleanup of cell debris–including removal of beads, dead cells, and doublets–and negative selection of CD45+ cells was performed (Supplementary Fig. 2G) using Cytobank software[114,115]. Visual representation of CD45- single-cell data was achieved using UMAP[116] (15 neighbors, 0.01 minimum distance, outliers collapsed), and viSNE mapping (5,000 iterations, 100 perplexity, 0.5 theta), the latter of which is based on the t-Distributed Stochastic Neighbor Embedding (t-SNE) algorithm[117]. FlowSOM clustering was performed within Cytobank (hierarchical consensus, 10 iterations) and cluster labels were assigned using established literature on skeletal cell types (Table 1), with relative marker intensities per cluster visualized by heatmap. FCS files were exported, concatenated in R, then re-uploaded for visualization of merged populations. Quantified values were exported to Graphpad Prism 8 to construct plots and perform statistical analyses.

**CITRUS analysis.** CITRUS analyses[118] were performed in Cytobank using the Significance Analysis of Microarrays (SAM) correlative association model. Nearest Shrunken Centroid (PAMR) and L1-Penalized Regression (LASSO via GLMNET) predictive association models were run simultaneously to analyze model error rates to confirm the validity of the statistical model. For the CITRUS assessment of median expression changes, cells were clustered by identification markers and statistics channels included all functional markers; for the assessment of abundances, all markers were used for clustering. All CITRUS analyses used the following settings: 2000 events samples per file, 2% minimum cluster size, 5 cross-validation folds, and 5% false discovery rate (FDR).

**Pseudotime.** Exported FCS files from Cytobank were imported into R using the CytoTree package (v1.0.3)[119]. Briefly, samples were merged per condition and clustered into minimum spanning trees for cell trajectory inference. Pseudotime calculation was performed by defining root clusters based on expression of stem cell markers (CD146, Sca-1, PDGFRα, Nestin, LeptinR). Representation of pseudotime was performed using diffusion mapping, along with density and trajectory plots.

## Fluorescence-assisted cell sorting (FACS)

Single-cell suspensions of mesenchymal skeletal cells were prepared, as described above. Samples were incubated with anti-mouse CD45-FITC at 1:400 dilution in FACS buffer at 4 °C for 20 min in the dark. Depending on the experiment, cells were also stained with TotalSeq-B0212 anti-mouse CD24 antibody (Biolegend) (CITE-Seq) or CD24-PECy7 (Biolegend) (CD24- and + cell isolation) at 1:400 dilution. Cells were then incubated with SYTOX blue at 1:4000 for 5 min, spun down at 300xg for 5 min at 4 °C, then resuspended in FACS buffer at $1 \times 10^7$ cells/mL and analyzed on a FACS Aria II (BD Biosciences). Unstained and single-color-stained controls were used for compensation and to control the gating strategy. Post-run flow cytometry data was analyzed and visualized with Cytobank software.

## CITE- and scRNA-seq library preparation

Live Lin-CD45- cells, isolated by FACS, were washed twice in 1x PBS + 0.04% BSA and immediately submitted to the Mayo Clinic Genome Analysis Core for single-cell sorting. The cells were counted and measured for viability using the Vi-Cell XR Cell Viability Analyzer (Beckman-Coulter). The Chromium Next GEM Single Cell 3' Library and Gel Bead Kit (10x Genomics) was used for cDNA synthesis and standard Illumina sequencing primers and a set of unique i7 Sample dual indices (10x Genomics) were added to each cDNA pool. All cDNA pools and resulting libraries were measured using Qubit High Sensitivity assays (Thermo Fisher Scientific) and Agilent Bioanalyzer High Sensitivity chips (Agilent). Pooled libraries were then sequenced on Illumina NextSeq 2000 at 40,000 and 50,000 fragment reads/samples following Illumina's standard protocol on a P2 flow cell. The P2 flow cell was sequenced at $100 \times 2$ paired-end reads using a NextSeq P2 sequencing kit, NextSeq Control Software v1.4.1 and base-calling was analyzed using Illumina's RTA version 3.4.4. 10X Genomics Cell Ranger Single Cell Software Suite (v6.0.0) was used to demultiplex raw base call (BCL) files generated from the sequencer into FASTQ files. The pipeline input FASTQ files for each sample to perform alignment to the reference genome, filtering, barcode counting, and UMI counting.

## CITE- and scRNA-seq analysis

Seurat package (v4.0)[120] was used in R to perform integrated analyses of single cells. Genes expressed in <3 cells and cells that expressed <200 genes and >20% mitochondria genes were excluded for downstream analysis in each sample. Each dataset was SCTransform-normalized and the top 3000 Highly Variable Genes (HVGs) across cells were selected. The datasets were integrated based on anchors identified between datasets before Principal Component Analysis (PCA) was performed for linear dimensional reduction. Shared Nearest Neighbor (SNN) Graph were constructed to identify clusters on the low-dimensional space (top 30 statistically significant principal components (PCs). Enriched marker genes in each cluster conserved across all samples were identified. An unbiased clustering according to the recommendations of the Seurat package was used with a resolution of 0.8. For Uniform Manifold Approximation and Projection for Dimension Reduction (UMAP) calculations, the RunUMAP function (dims = 1:40, reduction = "pca") was utilized, and both DimPlot (Seurat) and plot_density (Nebulosa) used for plotting. The CellChat[121,122] package was utilized for secretory signaling analyses.

## Immunofluorescent staining

Femurs from 6-month-old *Runx2-Cre x TdTomato* mice were fixed in 4% paraformaldehyde (PFA) at 4 °C for 72 h under gentle agitation. Bones were decalcified in 10% EDTA for 2 weeks at 4 °C under gentle shaking agitation, followed by incubation in 30% sucrose for 3 days at 4 °C. Samples were embedded in Cryomatrix (ThermoFisher Scientific, Wilmington, DE) and flash frozen in liquid nitrogen and stored at −80 °C. 7 µm-thick cryosections were prepared for fluorescent imaging. Bone sections were stained with anti-mouse CD24 antibody (Biolegend) at 1:50 dilution at 4 °C overnight, followed by anti-Rat-FITC secondary antibody at 1:200 for 1 h at room temperature. Slides were then mounted with ProLong Antifade (ThermoFisher) and imaged on an LSM 980 Airyscan 2 confocal microscope (Carl Zeiss Microscopy) at 63X magnification.

## In vitro senolysis assay

Single-cell suspensions of bone and marrow cells from 24-month-old INK-ATTAC mice (*n* = 4 males) were prepared as described above for CyTOF experiments, followed by subsequent depletion of hematopoietic (CD45) and immune (Lin) cells by MACS. Cells were plated in duplicate in 96-well low-adherence plates (Corning) and treated with either vehicle (EtOH) or AP20187 (1 µM) for 24 h at 37 °C. Cells were then washed with PBS and stained for 15 min in Annexin V Binding Buffer (Biolegend) in the dark at 4 °C with Annexin V-Alexa 647 (Biolegend) at a 1:20 dilution, CD24-PE-Cy7 at 1:400 dilution, and SyTOX Blue at 1:4000 dilution. Cells were then analyzed by flow cytometry, with apoptotic cells defined by percent Annexin V positive.

## Cell culture staining

**SA-β-Gal.** To assess senescence in vitro, cellular SA-β-Gal activity was measured as described previously[94] using the Cell Signaling Technology Senescence β-Galactosidase Staining Kit. Briefly, CD24+ or − cells were seeded on 8-well chamber slides at $1 \times 10^4$ cells/cm² and allowed to grow for 14 days. Cells were then washed in PBS (pH 7.4) and fixed with 1X fixative solution for 5 min, then washed three times using PBS. Cells were then incubated in 1X SA-β-Gal staining solution at 37 °C for 16 hr. Cells were washed in ice-cold PBS and mounted with DAPI Pro-Long (ThermoFisher) staining nuclei for cell counting. In blinded fashion, ten images per well were taken from random fields using fluorescence microscopy (Nikon Eclipse Ti) and SA-β-Gal-positive cells were counted and reported as a percentage of total cells.

**Crystal violet.** CD24- and + cells were seeded directly from FACS-mediated isolation into 25cm² flasks and allowed to grow for 7 days.

Cells were then washed in PBS and fixed in 4% PFA for 20 min. Fixed cells were washed again with PBS and stained with 1% crystal violet in 20 % ethanol for 20 min. Excess dye was removed by washing with distilled water (dH₂0) and images were acquired upon drying. Colony forming efficiency (CFE) was determined for each sample by counting colonies containing over 50 cells, then dividing by total sorted cells.

**Alizarin red and alkaline phosphatase.** Cells were washed with PBS then fixed with 4% PFA for 10 min. For Alkaline Phosphatase (ALP) analysis, the fixed cells were stained with 1-Step NBT/BCIP Substrate Solution (ThermoFisher) in the dark for 30 min. To detect mineralization, cells were fixed in 4% PFA and stained with Alizarin Red (Millipore) for 30 min; both stains were washed with dH₂0 and let dry before imaging.

## Quantitative real-time polymerase chain reaction (qPCR) analysis

Total RNA was extracted according to the manufacturer's instructions using QIAzol Lysis Reagent. Purification with RNeasy Mini Columns (QIAGEN, Valencia, CA) was subsequently performed. On-column RNase-free DNase solution (QIAGEN, Valencia, CA), was applied to degrade contaminating genomic DNA. RNA quantity was assessed with Nanodrop spectrophotometry (Thermo Fisher Scientific, Wilmington, DE). Standard reverse transcriptase was performed using High-Capacity cDNA Reverse Transcription Kit (Applied Biosystems by Life Technologies, Foster City, CA). Transcript mRNA levels were determined by qRT-PCR on the ABI Prism 7900HT Real-Time System (Applied Biosystems, Carlsbad, CA), using SYBR green (Qiagen, Valencia, CA). The mouse primer sequences, designed using Primer Express Software Version 3.0 (Applied Biosystems), for the genes measured by SYBR green are provided in Supplementary Table 2. Input RNA was normalized using two reference genes (*Actb, Gapdh*) from which the most stable reference gene was determined by the geNorm algorithm. For each sample, the median cycle threshold (Ct) of each gene (run in triplicate) was normalized to the geometric mean of the median Ct of the most stable reference gene. The delta Ct for each gene was used to calculate the relative mRNA expression changes for each sample. Genes with Ct values > 35 were considered not expressed (NE), as done previously[6].

## Statistics & reproducibility

In-text results and bar plots are mean ± standard deviation. Box plots show median and interquartile range, and error bars represent minimum and maximum values. Sample sizes were determined based on previously conducted and published experiments[16,94] in which statistically significant differences were observed among various senescence and skeletal parameters in response to aging or senolytic treatment. Animal numbers are indicated in the figure legends, and all samples presented represent biological replicates. No data were excluded from the analyses in this study. Mice undergoing treatment were randomized by weight before dosing. We did not exclude mice, samples, or data points from analyses. Non-Gaussian distributions were detected using the Shapiro-Wilk normality test. If the normality or equal variance assumptions for parametric analysis methods were not met (Shapiro-Wilk $p < 0.05$), data were analyzed using non-parametric tests (e.g., Mann−Whitney test). Otherwise, differences between groups were analyzed by parametric tests; figure legends indicate the statistical tests used in each experiment. Statistical analyses were performed using GraphPad Prism (Version 8.0) and R (v4.03). A $p$-value < 0.05 was considered statistically significant. Heatmap values were transformed by subtraction of row mean and dividing by standard deviation, visualized in Morpheus – Broad Institute. Experimental design diagrams and schematics were made using BioRender.com. Venn diagrams were made using Meta-Chart.com.

## Reporting summary

Further information on research design is available in the Nature Portfolio Reporting Summary linked to this article.

## Data availability

The CyTOF data generated in this study have been deposited in the Mendeley database under https://doi.org/10.17632/hg7sd7hbk5.2 [https://data.mendeley.com/datasets/hg7sd7hbk5/draft?a=3146d51a-81ab-4453-a602-79dc390f522e]. The scRNA-seq and CITE-seq data generated in this study have been deposited in the Gene Expression Omnibus (GEO) under accessions GSE237307 and GSE237301, respectively. Source data are provided with this paper.

## Code availability

CyTOF data was analyzed in software that does not require code (Cytobank). Both CITE-seq and scRNA-seq analyses were performed using standard Seurat and CellChat analysis pipelines, with specific code scripts available upon request.

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

## Acknowledgements

We would like to thank the Mayo Clinic Immune Monitoring Core, and specifically Samera Farwana, for the efforts in planning and running CyTOF samples. We would also like to thank the Mayo Clinic Genome Analysis Core and Co-Directors, Julie M. Cunningham, Ph.D., and Eric Wieben, Ph.D. in planning and running sequencing experiments. In addition, this work was supported by the Mayo Clinic Center for Clinical and Translational Science (CCaTS). This work was supported by the American Federation of Aging Research (AFAR) Glenn Foundation for Medical Research Postdoctoral Fellowship in Aging Research (M.L.D.), Robert and Arlene Kogod Center on Aging Pilot Funds (M.L.D.), Mayo Clinic Edward C. Kendall Fellowship (M.L.D.), the National Institutes of Health (NIH) grants UL1 TR002377 (M.L.D.), P01 AG062413 (S.K., D.G.M., J.N.F.), R01 AG076515 (S.K., D.G.M.), U54 AG079754 (S.K.), R01 AG063707 (D.G.M.), R01 DK128552 (J.N.F.), T32 AG049672 (M.L.D.), and the German Research Foundation (D.F.G., 413501650) (D.S.).

## Author contributions

M.L.D. and S.K. conceived and directed the project. M.L.D. and S.K. designed the experiments with input from K.P., D.G.M., and J.N.F. Experiments were performed by M.L.D., J.L.R., S.J.V., and J.K. Data was analyzed by M.L.D., and interpreted with input from S.K. and D.S. M.L.D. and S.K. wrote the manuscript, which all authors then reviewed. S.K. supervised all experimental design, data analyses, and manuscript preparation.

## Competing interests

The authors declare no competing interests.
