## [Peer Review File · Nature Communications]

Multiparametric senescent cell phenotyping reveals targets of senolytic therapy in the aged murine skeletonREVIEWER COMMENTS

Reviewer #1 (Remarks to the Author):

This is a strong descriptive, technically rigorous study investigating senescent bone marrow mesenchymal cells at single-cell resolution by CyTOF mass cytometry. Using carefully validated antibodies for canonical senescence markers p16 and p21, the authors first demonstrated that a subset of non-proliferating anti-apoptotic p16 cells, p16KB cells, were particularly senescent and increased with age. Second, they ablated p16 cells with inducible INK-ATTAC (“genetic senolytic clearance”) and found that CD24+ cells were cleared, which also occurred due to pharmacological senolysis by D+Q. CD24+ cells showed SASP and could not be cultured in vitro. From these data, the authors conclude that CD24 is a novel marker of senescent mesenchymal cells.

Overall, the study is well-designed, the manuscript is well-written, and the rationale is well-formulated. This study is innovative because the authors used a large panel of antibodies to comprehensively characterize senescent cells, instead of one or two markers as performed in previous studies. Extensive use of CyTOF with high technical rigor is also considered the strength of this study. This study has potentially strong impact on aging bone research. However, there are several key data that are missing from the current manuscript, which makes it somewhat difficult to correctly interpret the findings. My comments are listed below.

1. The authors should define more definitively the relationship between p16KB (or parental P16) cells and CD24 cells. The CyTOF plot in Figure 3F showed that inducible INK-ATTAC almost specifically ablated CD24+ cells and late osteoblasts/osteocytes, yet the authors did not demonstrate the distribution of p16 cells and its p16KB subset in the plot, in Figure 3A or elsewhere. Showing the distribution by a dot plot or a heat map would strengthen the link of the two cell types. This data will be important for readers to understand why p16-based cell ablation is highly specific to CD24+ cells.
2. The authors should validate the in vivo identity of CD24+ cells using histological approaches. CD24 antibodies appear to work well in sections. There is no definitive evidence that these CD24+ cells are mesenchymal. The current designation of CD24+ subsets is very confusing. For example, what is the difference between CD24+Runx2+Osterix+ and CD24+Osterix+ clusters? Does it mean that the latter cluster does not express Runx2? Without validating their cell identities in vivo, it is very difficult to make any inference on putative cell states of these cells, including senescence.
3. Related to the first point, is it possible to define the relationship between p16+ cells and CD24+ cells by staining CD24 in INKBRITE mice?
4. The authors should confirm if CD24+ cells are actually “cleared” due to apoptosis or other mechanisms in INK-ATTAC or D+Q. The authors convincingly showed that genetic and pharmacological senolytic clearance reduced the relative abundance of CD24+ cells and late osteoblasts/osteocytes and increased the relative abundance of early osteoblasts. The authors postulated that this is because these differentiated senescent cells were removed by senolysis. However, the alternative explanation for this shift in cell populations is that INK-ATTAC or D+Q blocked the terminal differentiation of early osteoblasts. It may have nothing to do with the clearance of differentiated cells. These cells are destined to spontaneously disappear or become embedded in matrix in a short amount of time anyway.
5. The cell culture data in Figure 6 appears to be overinterpreted. The authors showed in the preceding section that CD24+ cells are differentiated in nature, therefore it is fully expected that CD24+ cells do not proliferate well in vitro. The authors’ observation that FACS-sorted CD24+ cells did not proliferate and differentiate into Alizarin Red+ cells simply reflects the fact that these cells are already in a post-mitotic stage. These findings may have nothing to do with cell senescence.

Other points:

6. Line 111: I felt that it was a bit out of place to mention INK-ATTAC mice here, as the

authors did not actually use them until Line 278.

7. Line 157: The increase in BCL-2+ and BCL-2+/gH2A-X+ cells in Figure 2C appear to be very modest with no biological replicates included. Perhaps the authors should dampen the statement.

8. Line 181: Again, the increase of p21KB cells appear to be very modest.

9. Line 209: I am not sure if “near absence in youth” is an appropriate interpretation of the data. The bar plot seems to show that approximately 0.1% of BM mesenchymal cells are p16KB, which translates into as many as 1,000 cells per 1 million cells. Maybe it should be stated as “rare cells”.

10. Line 219: Skeletal stem cells should not be equated to BMSCs. Instead, BMSCs include a small population of skeletal stem cells.

11. Line 232: As mentioned above, the authors’ definition of CD24^{high} osteolineage cells (CD24+Runx2+Osterix+) and CD24+/Osterix+ cells is very confusing.

12. Line 265: The authors should clarify that “some markers were expressed by very low/undetectable levels [in scRNA-seq].

13. Line 285: As mentioned above, the caveat of this AP treatment is that it extends over 2 weeks (total of 4 treatment). This time frame is long enough for pre-existing osteoblasts to spontaneously undergo apoptosis or matrix embedding. So, it is still possible that the disappearance of CD24+ osteoblasts may be due to the blockage of differentiation from osteoblast precursors.

14. Line 378: The change in osteoblast/osteocyte clusters seems to be very modest, despite statistical significance.

15. Line 389: Again, the change in CD24+ cells appears to be very modest, despite statistical significance.

16. Line 426: As mentioned above, it appears that CD24+ cells have limited osteogenic potential because they are already differentiated and unable to proliferate well in culture.

17. Figure 2H: Could the authors specify in the main text that p16KB cells represent 17.4% of p16+ cells?

18. Figure 2I: More explanations would be needed for this key data. First, what does the x-axis show in the plot? Second, they should also include the plot of parental p16+ cells to emphasize the unique nature of p16KB cells. They should also point out that p16KB cells are highly heterogeneous in terms of SASP signatures.

19. Figure 3C: Can the authors explain a bit further why INK-ATTAC was specific to only 4 clusters? I wonder why other cell clusters with high p16 mean expression, such as ALPL+ and Sca1/PDGFRa, were not ablated in the genetic approach.

Reviewer #2 (Remarks to the Author):

In several previous manuscripts, this group has studied cellular senescence in bone. Notably, they recently used scRNAseq to identify a unique gene set that identifies senescence in hematopoietic and mesenchymal stem cells (PMID 35974106). The current study is designed to move this characterization from the genomic to the proteomic level; and to define a robust phenotype for senescence in osteolineage cells that will be helpful for future therapeutics. The first half of this study is focused on the enrichment of P16+Ki67-Bcl2+ expression in senescent mesenchymal stromal cells. The 2nd half was focused on identifying CD24 as a novel marker for osteolineage senescence. The work presented will be of interest to investigators in bone aging and senescence. In addition, the work could be of interest to hematologists, especially immunologists, due to the expression of CD24 on immune cells. However, the relevance of the work, especially the 2nd half pertaining CD24, would be dependent on the role/ mechanism of CD24 in senescent cells. The authors have acknowledged this in their discussion.

Major comments

1. In supplemental figure 2C and E, the FlowSOM plots of marrow only vs marrow+ bone digests indicated 3 clusters which were unique to the bone digests. This is verified by ALPL and OCN expression whose expression is limited to the unique clusters. Runx2 is expressed both in hematopoietic and osteolineage cells. Therefore, its expression tSNE plot in 2E is also verified. But the high expression of osterix in clusters from the marrow does not coincide with previous literature. Is the abundant expression of osterix in bone marrow cells normal? Or does it indicate an issue with the antibody? Also, there is lack of clarification about why the marrow and bone digestions were combined. Wouldn't using just the bone digests be better to analyze osteolineage cells? Especially since the percentage of mesenchymal stromal cells in the marrow is extremely low.
2. In table 1 (Pg 6) and all throughout the paper inflammatory cytokines and intracellular proteins such as TNF- α , MCP1, various interleukins were assessed through CyTOF to identify the SASP phenotype. However, in the methods section (Pg 38 and 39), I could not find the use of any secretion inhibitor such as Monensin or Brefeldin A to inhibit the secretion of these cytokines into the culture media. The use of these inhibitors is important for intracellular staining of CyTOF since secretion of these cytokines would skew the actual protein expression within the cell. Was such an inhibitor used?
3. In Fig 2E it was shown that only 7.6% of all cells showed P16 expression. However, in Fig 2D it was shown that the secretory cluster, TNF- α /IL-1 α cluster, Bcl-2/H2AX and the Bcl2 alone cluster all express P16. In addition, these 4 clusters make up at least 50% of all cells (Fig 2B). It would be very helpful to put together a ViSNE plot annotating P16 or Ki67 expression in all clusters, so that we can visually see this inverse correlation between the two markers.
4. Fig 3F and Supplemental Fig 5A. The difference in CD24 expression between the CD24 low osteolineage and CD24 high osteolineage cluster is not visible in the ViSNE plot. In fact, the cluster that was identified as CD24 low or negative (dark purple cluster in SFig 5A and main Fig 3F) still have a good amount of CD24 protein expression seen in red in the ViSNE plot in Supplemental 5A right corner. This same CD24 low purple cluster increased in cell numbers on application of the AP senolytic. If senolytics targeting CD24 expression were made, would they also kill this good "CD24 low" cluster which helped recover osteolineage cells from senescence? This would be a key question for which the actual function of CD24 in senescence needs to be determined. Also, to answer this, more characterization of heterogeneity within CD24 cells would be required.
5. Pg 21, line 381 "In sequential analyses, we found that D+Q reduced expression of p16, SASP factors PAI-1 and IL-1 β , and the DNA damage marker pATM within CD24/Runx2/Osterix+ cells (Fig. 5D; Extended Data Fig. 8C)." Please provide the data for these markers in 5D like it was shown in 5C for Runx2, osterix and CD24 change in expression. Currently, 5D shows the summary of the changes in protein expression for PAI-1 and IL-1 β , and the DNA damage marker pATM, but it does not show the actual expression change.

Minor comments

1. The CyTOF dot plots in supplemental figure 1 and 2 are not as clear when zoomed in.
2. Pg11 Fig 2B: Equal number of cells between the young and old FlowSOM plots would make this sub figure visually more convincing.
3. The number of times each CyTOF experiment was repeated has not been mentioned. Was it a minimum of N=3?
4. The number of mice used in figure 2 has not been mentioned.
5. Pg13, line 226: Full form of PDPN
6. It would be very helpful if supplemental figure 5B ViSNE plot of CD24 be duplicated or mentioned in text next to Fig 3F.
7. Pg18 line 337 the figures are labeled as 5C and D when they are 4C and D.
8. Supplemental figure legend 8 line 82: The gating strategy is labeled as C instead of D.

Reviewer #3 (Remarks to the Author):

Doolittle et al tried to characterize senescent cells in the skeleton using mass cytometry by time-of-flight (CyTOF). They used p16+/Ki67-/BCL-2+ (p16KB⁺) to define senescent cells and analyzed them at the single cell level. They further show that p16KB cell percentages were increased in CD24+ osteolineage cells, which were robustly cleared by both genetic and pharmacologic senolytic therapies in aged mice. CD24+ skeletal cells exhibited growth arrest, SA- β gal positivity, and impaired osteogenesis in vitro. They conclude that their studies provide a new approach to define senescent mesenchymal cells in vivo and identify a senescent CD24+ osteolineage population cleared by senolytics. Overall, the study provides very valuable information regarding bone aging and senile osteoporosis development. However, the following concerns need to be addressed before publication.

Major concerns:

1. Characterization of p16KB cells. While the authors used scRNA-seq data to verify the clustering of p16KB cells, they only used 3362 cells. This is not enough. Moreover, cell surface markers for mouse osteogenic lineage cells (from SSCs to osteoblasts) have been reported (Chan CK, Cell, 2015), which can also be used to define the 16KB cells. Lastly, immunostaining of bone sections is required to locate the p16KB cells.
2. The identity of CD24+ cells. If the Cre or CreERT mice are not available, cell surface marker expression can be used (Chan CK, Cell, 2015). There is a recent report by Jeroen van de Peppel (Stem Cells Dev 2021 30(6):325-336. Cell Surface Glycoprotein CD24 Marks Bone Marrow-Derived Human Mesenchymal Stem/Stromal Cells with Reduced Proliferative and Differentiation Capacity In Vitro), which may compromise the novelty of this manuscript.

Minor comments:

1. Can the author explain why D+Q cannot kill most of the p16KB cells. Why are the CD24+ cells so special?
2. CD24 has become a very interesting marker in cancer and cancer immunotherapy. What may have stimulated CD24 expression in the osteogenic cells?
3. I am not sure that etoposide is a proper inducer of p16 as it is a genotoxic drug that activates the p53 pathway.

Response to Reviewers

Overall response to Reviewers

We thank the Reviewers for their careful consideration of our manuscript. We have carefully considered each of the points raised and have either modified the paper, including additional experiments as appropriate, or provided a reason for not being able to fully address the specific point. We should note that we have not only re-analyzed existing data, but also performed substantial new experiments, including histological analyses for CD24+ cells, CITE-Seq with the CD24 antibody, and *in vitro* studies demonstrating induction of apoptosis by AP20187 in CD24+ cells, but not in CD24- cells, from *INK-ATTAC* mice. All changes to the original manuscript are indicated by tracking. Where relevant, for some of the responses, we are including Figures in this response letter as well as in the manuscript/Extended data to reduce the burden on the Reviewers. All page numbers in the response letter refer to the “clean” version of the revised manuscript.

Reviewer #1

This is a strong descriptive, technically rigorous study investigating senescent bone marrow mesenchymal cells at single-cell resolution by CyTOF mass cytometry. Using carefully validated antibodies for canonical senescence markers p16 and p21, the authors first demonstrated that a subset of non-proliferating anti-apoptotic p16 cells, p16KB cells, were particularly senescent and increased with age. Second, they ablated p16 cells with inducible INK-ATTAC (“genetic senolytic clearance”) and found that CD24+ cells were cleared, which also occurred due to pharmacological senolysis by D+Q. CD24+ cells showed SASP and could not be cultured in vitro. From these data, the authors conclude that CD24 is a novel marker of senescent mesenchymal cells.

Overall, the study is well-designed, the manuscript is well-written, and the rationale is well-formulated. This study is innovative because the authors used a large panel of antibodies to comprehensively characterize senescent cells, instead of one or two markers as performed in previous studies. Extensive use of CyTOF with high technical rigor is also considered the strength of this study. This study has potentially strong impact on aging bone research. However, there are several key data that are missing from the current manuscript, which makes it somewhat difficult to correctly interpret the findings. My comments are listed below.

Response: We thank the reviewer for the positive comments regarding our work.

1. The authors should define more definitively the relationship between p16KB (or parental P16) cells and CD24 cells. The CyTOF plot in Figure 3F showed that inducible INK-ATTAC almost specifically ablated CD24+ cells and late osteoblasts/osteocytes, yet the authors did not demonstrate the distribution of p16 cells and its p16KB subset in the plot, in Figure 3A or elsewhere. Showing the distribution by a dot plot or a heat map would strengthen the link of the two cell types. This data will be important for readers to understand why p16-based cell ablation is highly specific to CD24+ cells.

Response: Thank you for bringing up this important issue. In the revision, we have now specifically linked p16 expression to the CD24+ cells and their clearance by AP20187 in the *INK-ATTAC* mice. Note that for this analysis, we focused on the parental p16+ cells, as the transgene is driven by the p16 promoter and thus potentially targets all p16+ cells. To do so, we took advantage of the fact that in the *INK-ATTAC* transgene, the caspase 8 driven by the p16 promoter is tagged with a FLAG sequence, allowing for the demonstration of caspase 8/FLAG expression and also for monitoring the activity of the p16 promoter in the transgene (which is a 2.6 kB fragment of the endogenous p16 promoter).^{2, 3} Extended Data Fig. 8D, E (reproduced below)

overlays p16 protein expression on the cell populations shown in Fig. 3A and F, and Extended Data Fig. 8F overlays FLAG expression (reflecting the transgene promoter fragment). As is evident, there is reassuring concordance between the endogenous p16 protein (Extended Data Fig. 8E) and the caspase 8/FLAG protein produced by the transgene promoter fragment (Extended Data Fig. 8F). Moreover, the 2 populations most robustly cleared by AP in the INK-ATTAC mice – the CD24^{high} osteolineage cells and the late osteoblasts/osteocytes (Fig. 3F, G) also exhibit high p16 and caspase 8/FLAG expression (arrows in Extended Data Fig. 8E, F). There is modest clearance of CD24⁺/Osterix⁺ cells (Fig. 3G), and although these cells do not have particularly high p16 and caspase 8/FLAG levels, clearly those levels are sufficient to reduce their numbers following AP treatment. We acknowledge, however, that there do appear to be some populations (e.g., early osteoblasts,) where sub-populations of cells have high p16/caspase 8/FLAG expression, but do not appear to be cleared by AP. The reasons for this are unclear at present, but it is possible that these cells are cleared but rapidly repopulate, as the mice are harvested 4-6 days following the last AP dose. Further studies are needed, however, to address this issue.

2. *The authors should validate the in vivo identity of CD24⁺ cells using histological approaches. CD24 antibodies appear to work well in sections. There is no definitive evidence that these CD24⁺ cells are mesenchymal. The current designation of CD24⁺ subsets is very confusing. For example, what is the difference between CD24⁺Runx2⁺Osterix⁺ and CD24⁺Osterix⁺ clusters? Does it mean that the latter cluster does not express Runx2? Without validating their cell identities in vivo, it is very difficult to make any inference on putative cell states of these cells, including senescence.*

Response: This is a valid point particularly because, as compared to Runx2 and Osterix, CD24 has a less clearly defined role in mesenchymal cells within the bone microenvironment and is also expressed on some immune cells.⁴ First, as recommended, we performed histological staining of CD24 on bone sections. We found that a subset of CD24 cells do indeed co-express *Runx2*, as observed through CD24 staining on bone sections from *Runx2-Tdtomato* reporter mice (Extended Data Fig 5C). These CD24⁺Runx2⁺ cells exist at the bone surface and exhibit a morphology consistent with osteoblasts and/or bone lining cells. These data support our classification of our CyTOF CD24⁺ populations as osteolineage cells.

Furthermore, we performed CITE-Seq with the CD24 antibody on mesenchymal bone/marrow cell suspensions that were prepared and purified in a similar manner to our CyTOF cell isolations (Lin-CD45⁻). We found that CD24 marks a population that co-expresses *Runx2* and *Sp7* (Osterix) and cells expressing all three markers designate a distinct osteogenic population (Extended Data Fig 5D). We found a segregation of CD24⁺Runx2⁺Sp7⁺ and CD24⁺Sp7⁺ populations, finding that the CD24⁺Sp7⁺ subpopulation co-expresses late osteogenic markers (e.g., *Alpl*, *Bglap*) (Extended Data Fig 5E, F). These populations likely represent the clusters we observe within our CyTOF data and help to explain our designation of CD24⁺ subsets, which was based on their

expression in the heatmap (Fig 3B). We have placed this data and results in the manuscript and have also attempted to clarify this point further on page 13, where we note that “the latter cells emerged as a distinct cluster and had lower expression of Runx2 as well as somewhat lower expression of Osterix than the CD24^{high/low} osteolineage (CD24+/Runx2+/Osterix+) cells (Figure 3B).” These clusters emerged from the CyTOF analysis, and we acknowledge that within the scope of this paper, we cannot fully validate all cell identities, although in response to the first part of this point, we would submit that we have made a substantial effort to provide a detailed characterization of the CD24+ osteogenic cells.

3. Related to the first point, is it possible to define the relationship between p16+ cells and CD24+ cells by staining CD24 in INKBRITE mice?

Response: This is an excellent suggestion. We have very recently obtained the INK-BRITE mice from Dr. Peng at UCSF and are in the process of aging them. However, that will take over 2 years, so we would plan to perform these studies in the future, and this would be beyond the scope of the present work.

4. The authors should confirm if CD24+ cells are actually “cleared” due to apoptosis or other mechanisms in INK-ATTAC or D+Q. The authors convincingly showed that genetic and pharmacological senolytic clearance reduced the relative abundance of CD24+ cells and late osteoblasts/osteocytes and increased the relative abundance of early osteoblasts. The authors postulated that this is because these differentiated senescent cells were removed by senolysis. However, the alternative explanation for this shift in cell populations is that INK-ATTAC or D+Q blocked the terminal differentiation of early osteoblasts. It may have nothing to do with the clearance of differentiated cells. These cells are destined to spontaneously disappear or become embedded in matrix in a short amount of time anyway.

Response: The reviewer brings up an important point, as alterations in osteogenic cell abundances could arise from a number of mechanisms. To address this concern, we have performed an *ex vivo* senolysis assay to test if CD24+ cells are truly targeted by genetic senolytic treatment. Briefly, we isolated skeletal mesenchymal cells from 24-month-old INK-ATTAC mice in a similar manner to our CyTOF preparations (CD45-Lin-) and treated with AP20187 *in vitro* for 24 hours. We found increased apoptosis in CD24+ cells, demonstrated by increased % Annexin V+ cells by flow cytometry, with no such change in CD24- cells (Extended Data 8G and reproduced below), providing further evidence that this population is targeted in the bone microenvironment of INK-ATTAC mice.

With regard to possible non-senescence effects of AP on osteogenesis, we have previously shown that similar AP treatment in young (12-month-old) INK-ATTAC mice does not influence skeletal phenotypes⁵, which suggests that AP likely does not have effects on osteogenic

populations outside of promoting the clearance of senescent cells, which are present in low numbers in young mice and hence AP does not have significant effects in the young mice.

5. *The cell culture data in Figure 6 appears to be overinterpreted. The authors showed in the preceding section that CD24+ cells are differentiated in nature, therefore it is fully expected that CD24+ cells do not proliferate well in vitro. The authors' observation that FACS-sorted CD24+ cells did not proliferate and differentiate into Alizarin Red+ cells simply reflects the fact that these cells are already in a post-mitotic stage. These findings may have nothing to do with cell senescence.*

Response: The reviewer raises a good point, and in response, we now note on page 26 “We acknowledge that the impaired proliferation of the CD24+ cells could be due to their differentiation status. However, these cells also exhibited impaired osteogenic capabilities and were highly positive for the senescence marker, SA-β-gal.”

Other points:

6. *Line 111: I felt that it was a bit out of place to mention INK-ATTAC mice here, as the authors did not actually use them until Line 278.*

Response: We introduced the *INK-ATTAC* mice at this point because we wanted to make the point that using this line throughout allowed us to compare cellular alterations resulting from both aging and senolytic clearance within the same mouse strain. We tried a few different approaches that would allow us to make this point and yet introduce these mice later, as suggested by the Reviewer, but could not really find one that worked, so have left this as is.

7. *Line 157: The increase in BCL-2+ and BCL-2+/gH2A-X+ cells in Figure 2C appear to be very modest with no biological replicates included. Perhaps the authors should dampen the statement.*

Response: We have dampened the statement to now read, “The BCL-2+ and BCL-2+/γH2A-X+ populations showed an increased trend with age...”

8. *Line 181: Again, the increase of p21KB cells appear to be very modest.*

Response: We have modified this statement to read, “Interestingly, despite no age-related increase in total p21+ cells, p21KB cells did increase significantly, albeit modestly, in old mice...”

9. *Line 209: I am not sure if “near absence in youth” is an appropriate interpretation of the data. The bar plot seems to show that approximately 0.1% of BM mesenchymal cells are p16KB, which translates into as many as 1,000 cells per 1 million cells. Maybe it should be stated as “rare cells”.*

Response: We have modified this statement to read, “...and a relatively low abundance in youth with a marked increase with aging.”

10. *Line 219: Skeletal stem cells should not be equated to BMSCs. Instead, BMSCs include a small population of skeletal stem cells.*

Response: Good point. We have removed the reference to SSCs.

11. *Line 232: As mentioned above, the authors' definition of CD24^{high} osteolineage cells (CD24+Runx2+Osterix+) and CD24+/Osterix+ cells is very confusing.*

Response: Please see our response to this issue in point 2, above.

12. *Line 265: The authors should clarify that “some markers were expressed by very low/undetectable levels [in scRNA-seq].*

Response: We agree this wording needs clarification. In our scRNA-seq data, *Sost* mRNA expression was not detected in any cell, while *Dmp1* and *Sp7* were detected, but at low levels. This is consistent with other studies, demonstrating low *Sost* mRNA expression in scRNA-seq

even in osteocyte-enriched samples⁶, whereas CyTOF detection of Sclerostin expression at the protein level was clear. This statement has been clarified in the text. To address this issue, we sequenced cells using CITE-seq on a more powerful sequencer (Extended Data Fig 5D-F), which gave better sequencing depth and allowed for further investigation into Sp7-expressing cells.

13. Line 285: As mentioned above, the caveat of this AP treatment is that it extends over 2 weeks (total of 4 treatment). This time frame is long enough for pre-existing osteoblasts to spontaneously undergo apoptosis or matrix embedding. So, it is still possible that the disappearance of CD24+ osteoblasts may be due to the blockage of differentiation from osteoblast precursors.

Response: See response to Point 4 with *in vitro* senolysis assay, demonstrating specificity of p16+ genetic cell clearance for CD24+ cells.

14. Line 378: The change in osteoblast/osteocyte clusters seems to be very modest, despite statistical significance.

Response: We have modified the statement in the text to read "...and late osteoblast/osteocyte clusters, although the reduction in the latter cells was modest."

15. Line 389: Again, the change in CD24+ cells appears to be very modest, despite statistical significance.

Response: The reduction in CD45-CD24+ cells in the INK-ATTAC model was ~50% (Fig. 5F) but admittedly more modest following DQ treatment. As such, we have modified the text to read, "...we found that total CD45-CD24+ stromal cells were reduced in both genetic (INK-ATTAC) and more modestly by pharmacological (D+Q) methods of senolytic clearance...".

16. Line 426: As mentioned above, it appears that CD24+ cells have limited osteogenic potential because they are already differentiated and unable to proliferate well in culture.

Response: Please see response to point 5, above.

17. Figure 2H: Could the authors specify in the main text that p16KB cells represent 17.4% of p16+ cells?

Response: This has been added to the text on page 10.

18. Figure 2I: More explanations would be needed for this key data. First, what does the x-axis show in the plot? Second, they should also include the plot of parental p16+ cells to emphasize the unique nature of p16KB cells. They should also point out that p16KB cells are highly heterogeneous in terms of SASP signatures.

Response: The x-axis are the 12 biological replicates (individual mice) in the Old group, side-by-side and separated by cell populations (p16- vs p16KB); this has been clarified in the figure legend. In response to the second point, we have generated a heatmap of p16-, p16+, and p16KB cells and placed in Extended Data Fig. 3D, which demonstrates higher expression levels of markers for both senescence (p21, p53) and SASP (TNF α , IL-6, IL-1 α , pNFkB, CXCL1) in the p16KB cells versus p16+ parental cells. Note that as compared to p16- cells, p16+ cells overall had higher expression of senescence and SASP markers, suggesting that these cells may be "pre-senescent" or at the least highly inflammatory, and further studies are needed to more fully distinguish the p16+ vs the p16KB cells. We have also pointed out the inter-animal heterogeneity of p16KB cells in the text.

19. Figure 3C: Can the authors explain a bit further why INK-ATTAC was specific to only 4 clusters? I wonder why other cell clusters with high p16 mean expression, such as ALPL+ and Sca1/PDGFR α , were not ablated in the genetic approach.

Response: The Reviewer raises a good point. We cannot provide a definitive answer at present, but the detailed response and edits to point 1, above, reflects our attempt to address this issue, where we also acknowledge that further studies are needed to answer this question.

Reviewer #2

In several previous manuscripts, this group has studied cellular senescence in bone. Notably, they recently used scRNAseq to identify a unique gene set that identifies senescence in hematopoietic and mesenchymal stem cells (PMID 35974106). The current study is designed to move this characterization from the genomic to the proteomic level; and to define a robust phenotype for senescence in osteolineage cells that will be helpful for future therapeutics. The first half of this study is focused on the enrichment of P16+Ki67-Bcl2+ expression in senescent mesenchymal stromal cells. The 2nd half was focused on identifying CD24 as a novel marker for osteolineage senescence. The work presented will be of interest to investigators in bone aging and senescence. In addition, the work could be of interest to hematologists, especially immunologists, due to the expression of CD24 on immune cells. However, the relevance of the work, especially the 2nd half pertaining CD24, would be dependent on the role/ mechanism of CD24 in senescent cells. The authors have acknowledged this in their discussion.

Response: We thank the Reviewer for the positive comments regarding our current and previous work.

Major comments

1. In supplemental figure 2C and E, the FlowSOM plots of marrow only vs marrow+ bone digests indicated 3 clusters which were unique to the bone digests. This is verified by ALPL and OCN expression whose expression is limited to the unique clusters. Runx2 is expressed both in hematopoietic and osteolineage cells. Therefore, its expression tSNE plot in 2E is also verified. But the high expression of osterix in clusters from the marrow does not coincide with previous literature. Is the abundant expression of osterix in bone marrow cells normal? Or does it indicate an issue with the antibody? Also, there is lack of clarification about why the marrow and bone digestions were combined. Wouldn't using just the bone digests be better to analyze osteolineage cells? Especially since the percentage of mesenchymal stromal cells in the marrow is extremely low.

Response: Thank you for the comment. We would like to first clarify that this sample was depleted of hematopoietic cells before analysis (Lin- [CD5, CD45R (B220), CD11b, Gr-1 (Ly-6G/C), Ly6B.2 and Ter-119] and CD45- cells) and thus all cells depicted in our FlowSOM plot in Extended Data Figure 2C and E are of mesenchymal origin, differing only by anatomical location (marrow only vs. marrow + bone). With regards to the issue of osterix expression in the marrow cells, we note that our protocol involved flushing of the marrow, which likely released at bone surface cells

expressing osterix as part of the “marrow” fraction. This has been observed by others in scRNA-seq⁷, as shown below (repurposed from Baryawno et al. *Cell* 2019), where Sp7 (Osterix) is expressed in cells derived from both bone and bone marrow (BM-arrows). Moreover, our osterix antibody (ThermoFisher cat# PA5-40411) has been shown by others to positively stain mouse calvarial osteoblasts⁸, supporting the specificity of this antibody in our study.

In terms of combining bone and marrow digestions, our goal was to comprehensively study skeletal lineage cells across aging, and thus we wanted to capture mesenchymal stem cells (e.g., BMSCs) in addition to our bone-embedded osteolineage populations. To accomplish this, we built upon a protocol extensively validated and published by the Scadden laboratory^{7,9}. While it is true that mesenchymal stromal cells in the marrow are rare, we enriched for these cells using the methods listed above (depletion of Lin+ and CD45+ hematopoietic cells). Similar enrichment of these cells has been successfully shown by others at the single-cell level^{7,9}. We have now clarified our rationale for this approach on page 7.

2. In table 1 (Pg 6) and all throughout the paper inflammatory cytokines and intracellular proteins such as TNF- α , MCP1, various interleukins were assessed through CyTOF to identify the SASP phenotype. However, in the methods section (Pg 38 and 39), I could not find the use of any secretion inhibitor such as Monensin or Brefeldin A to inhibit the secretion of these cytokines into the culture media. The use of these inhibitors is important for intracellular staining of CyTOF since secretion of these cytokines would skew the actual protein expression within the cell. Was such an inhibitor used?

Response: We have had extensive discussions with our CyTOF Core Laboratory both before the initiation of our studies and in order to respond to the Reviewer. Based on the considerations below, our CyTOF laboratory does not generally recommend the use of secretion inhibitors, so they were not used in our study, nor were they used in previous, similar CyTOF studies focusing on mesenchymal cells by others at Mayo¹⁰⁻¹² and elsewhere^{9, 13}. In general, these secretion inhibitors have been used on PBMCs following an activation stimulus *ex vivo* (e.g., PMA, LPS, etc. on cultured cells) in order to perform functional assays in culture¹⁴. First, rather than remaining in suspension *ex vivo* for any length of time, the CyTOF protocol involves fixing the cells soon after they are isolated, aiming to capture their baseline state *in vivo* as closely as possible. Second, both of these agents have effects beyond inhibition of cytokine secretion. Specifically, Monensin has been shown to impair viability and induce oxidative stress^{15, 16}, while Brefeldin A can alter cell-surface marker expression¹⁷. Third, our laboratory has found that our CyTOF method is sufficiently sensitive that an inhibitor is generally not needed in order to detect cytokines that are being secreted in the basal state. For example, without the use of these inhibitors, we were able to show that cells treated with the senescence inducer, etoposide, had marked upregulation of secreted SASP factors (Fig. 1D-F). The success of this pilot study justified application *in vivo*, and we indeed found that senescent (p16KB) cells were highly enriched for SASP factors as compared to non-senescent (p16-) cells (Fig. 2I), as anticipated.

3. In Fig 2E it was shown that only 7.6% of all cells showed P16 expression. However, in Fig 2D it was shown that the secretory cluster, TNF- α /IL-1 α cluster, Bcl-2/H2AX and the Bcl2 alone cluster all express P16. In addition, these 4 clusters make up at least 50% of all cells (Fig 2B). It would be very helpful to put together a VisNE plot annotating P16 or Ki67 expression in all clusters, so that we can visually see this inverse correlation between the two markers.

Response: We would like to clarify that the clusters analyzed in Fig. 2A-D are on p16+ cells only, and thus all clusters would have high levels of p16. The 7.6% p16+ mentioned (Fig 2E) is on all cells. A main finding of this work was that although p16 and Ki67 are generally inversely correlated (Fig. 2F, Extended Data Fig 3B) – which has also recently been shown by others¹⁸ – it is the co-expression of Bcl2 with p16 that is required to identify fully growth-arrested senescent cells. As shown below, in p16+ cells, cells expressing Bcl2 are negative for Ki67, and vice-versa, demonstrating the inverse relationship we uncovered between Ki67 and Bcl2.

4. Fig 3F and Supplemental Fig 5A. The difference in CD24 expression between the CD24 low osteolineage and CD24 high osteolineage cluster is not visible in the ViSNE plot. In fact, the cluster that was identified as CD24 low or negative (dark purple cluster in Sfig 5A and main Fig 3F) still have a good amount of CD24 protein expression seen in red in the ViSNE plot in Supplemental 5A right corner. This same CD24 low purple cluster increased in cell numbers on application of the AP senolytic. If senolytics targeting CD24 expression were made, would they also kill this good “CD24 low” cluster which helped recover osteolineage cells from senescence? This would be a key question for which the actual function of CD24 in senescence needs to be determined. Also, to answer this, more characterization of heterogeneity within CD24 cells would be required.

Response: This is an important point. Upon closer inspection, we found that the CD24^{low} Osteolineage cluster exhibits a higher percentage of Ki67⁺ cells (see below and in Extended Data Fig. 8H), which suggests it is a proliferative cellular population and may explain its increased prevalence after AP treatment. The classification of this cluster was based on mean CD24 expression, as shown by the heatmap. However, we acknowledge that these cells are likely functionally distinct from CD24^{high} osteolineage and CD24⁺Osterix⁺ clusters due to their dissimilarity in response to aging and senolytic treatment. We now address this in the Discussion on page 31: “We acknowledge, however, that additional studies are needed to evaluate the potential role of CD24 in marking senescent cells in other tissues beyond the bone microenvironment and across species. Moreover, senolytics potentially targeting CD24-expressing cells would require careful evaluation as at least non-senescent CD24⁺ cells may also have beneficial effects across tissues.”

5. Pg 21, line 381 “In sequential analyses, we found that D+Q reduced expression of p16, SASP factors PAI-1 and IL-1 β , and the DNA damage marker pATM within CD24/Runx2/Osterix⁺ cells (Fig. 5D; Extended Data Fig. 8C).” Please provide the data for these markers in 5D like it was shown in 5C for Runx2, osterix and CD24 change in expression. Currently, 5D shows the

summary of the changes in protein expression for PAI-1 and IL-1 β , and the DNA damage marker pATM, but it does not show the actual expression change.

Response: Thank you for the comment. We have created box plots of expression changes within the CD24/Runx2/Osterix+ clusters and placed them as callouts in Figure 5D. We decided on the box plots over the histograms (used in 5C) to visualize biological replicates within groups.

Minor comments

1. The CyTOF dot plots in supplemental figure 1 and 2 are not as clear when zoomed in.

Response: We apologize, and have rectified the dot plots in supplemental figures 1 and 2. The embedded figures in our PDF appear to have compromised image quality due to processing, but we will ensure the quality of images in each figure in the case of final publication.

2. Pg11 Fig 2B: Equal number of cells between the young and old FlowSOM plots would make this sub figure visually more convincing.

Response: While we agree it may be interesting to view in that manner, we justified using total cell amounts due to the low abundance of p16+ cells, which (as observed in the Young sample) can make it difficult to observe overall population changes through UMAP visualization due to scarcity of points. To exemplify, we have shown the plots with equal numbers below (842 cells):

3. The number of times each CyTOF experiment was repeated has not been mentioned. Was it a minimum of N=3?

Response: All samples were analyzed on freshly isolated cells across several collections, and thus the sample size (n) for CyTOF experiments in this was based on biological replicates, rather than repeated experiments (as would be done for cell culture). Each CyTOF run was typically performed on 3-4 mice at a time, and thus at least 3 runs per cohort were performed.

4. The number of mice used in figure 2 has not been mentioned.

Response: The analyses in Figure 2 were a continuum of those in Figure 1, and thus contained the same mice in each group (n=15 young, n=12 old). Individual replicates were combined for analyses in Fig. 2B-D due to low p16+ cell numbers per mouse. This has been clarified in the figure legend.

5. Pg13, line 226: Full form of PDPN.

Response: Correction made.

6. It would be very helpful if supplemental figure 5B ViSNE plot of CD24 be duplicated or mentioned in text next to Fig 3F.

Response: We agree with this point and have placed a CD24 viSNE expression plot in Fig 3F.

7. Pg18 line 337 the figures are labeled as 5C and D when they are 4C and D.

Response: Thank you for pointing out this error – correction made.

8. Supplemental figure legend 8 line 82: The gating strategy is labeled as C instead of D.

Response: Thank you for this correction, we have updated this figure legend accordingly.

Reviewer #3

Doolittle et al tried to characterize senescent cells in the skeleton using mass cytometry by time-of-flight (CyTOF). They used p16+/Ki67-/BCL-2+ (p16KB⁺) to define senescent cells and analyzed them at the single cell level. They further show that p16KB⁺ cell percentages were increased in CD24⁺ osteolineage cells, which were robustly cleared by both genetic and pharmacologic senolytic therapies in aged mice. CD24⁺ skeletal cells exhibited growth arrest, SA-β gal positivity, and impaired osteogenesis in vitro. They conclude that their studies provide a new approach to define senescent mesenchymal cells in vivo and identify a senescent CD24⁺ osteolineage population cleared by senolytics. Overall, the study provides very valuable information regarding bone aging and senile osteoporosis development. However, the following concerns need to be addressed before publication.

Response: We thank the Reviewer for the positive comments regarding our work.

Major concerns:

1. *Characterization of p16KB cells. While the authors used scRNA-seq data to verify the clustering of p16KB cells, they only used 3362 cells. This is not enough. Moreover, cells surface markers for mouse osteogenic lineage cells (from SSCs to osteoblasts) have been reported (Chan CK, Cell, 2015), which can also be used to define the 16KB cells. Lastly, immunostaining of bone sections is required to locate the p16KB cells.*

Response: Thank you for this comment. To build upon our existing scRNA-seq analyses, we performed a deeper analysis of p16KB cells on our newly generated CITE-Seq data of the same cell preparation. As noted in the response to Reviewer 1, point 2, we performed CITE-Seq with the CD24 antibody on mesenchymal bone/marrow cell suspensions that were prepared and purified in a similar manner to our CyTOF cell isolations (Lin-CD45⁻). In this analysis, we were able to identify *Cdkn2a*⁺ (p16⁺) cells which contained mutually exclusive *Mki67*⁺ or *Bcl2*⁺ subpopulations (Extended Data Fig 3E, F). The p16KB (*Cdkn2a*⁺*Mki67*⁻*Bcl2*⁻) population (Cluster 4 in Extended Data Fig 3E) demonstrated robust co-expression of other well-characterized anti-apoptosis genes including *Bcl2l2* (BCL-W)¹⁹, *Mcl1*^{20, 21}, *Xiap*²², *Birc6*²³, as well as *Mapk14*, which encodes the major subunit (p38α) in p38MAPK; a robust regulator of senescence and SASP²⁴⁻²⁶ (Extended Data Fig 3G). To further investigate the SASP in these cells, we performed CellChat analysis and found that the p16KB cell cluster exhibited the highest levels of outgoing secretory signals of any *Cdkn2a*⁺ population (Extended Data Fig 3H-J). We then applied our recently established senescence gene panel (“SenMayo”)²⁷ and indeed found the highest enrichment in p16KB cells (Extended Data Fig 3K, L).

We also investigated cell surface markers detailed in the referenced publication (Chan et al. Cell 2015) and found that while cluster 4 did not clearly overlap with any of the defined subtypes, it did resemble a sorting pattern for bone cartilage and stromal progenitors (BCSPs; CD45⁻Ter119⁻Tie2⁻CD51⁺Thy-6C3⁻CD105⁺) (see figure below). The non-inclusion of these markers in our CyTOF panel was a limitation of this study, which we addressed on page 32, citing this

important publication. We are working on future experiments to further characterize the identity of p16KB cells, which present with challenges due to their low abundance. However, from our large cohort CyTOF analyses using other established markers to differentiate between stem-like (Sca-1, Lepr, CD200, CD29) and differentiated (Runx2, Osterix, Alpl, Sclerostin) skeletal cell types, we observed a strong preference of p16KB cells (and senolytic specificity) to committed cell types (CD24^{high} osteolineage cells, Late Osteoblast/Osteocytes).

With regards to histological staining of p16KB cells, this remains challenging for several reasons. First, although we carefully validated the mouse p16 antibody used in the present studies for CyTOF, application of this antibody for histological analyses will require considerable further validation that is ongoing in our laboratory. Second, although we were easily able to detect p16KB cells using the high sensitivity of CyTOF, this did require a high degree of enrichment for mesenchymal cells, which is not feasible in histological analyses. Thus, although we agree that the histological identification of p16KB cells is very much worth pursuing, this is technically challenging and beyond what we can accomplish in a reasonable time-frame and would thus defer this to future studies.

2. The identity of CD24⁺ cells. If the Cre or CreERT mice are not available, cell surface marker expression can be used (Chan CK, Cell, 2015). There is a recent report by Jeroen van de Peppel (Stem Cells Dev 2021 30(6):325-336. Cell Surface Glycoprotein CD24 Marks Bone Marrow-Derived Human Mesenchymal Stem/Stromal Cells with Reduced Proliferative and Differentiation Capacity In Vitro), which may compromise the novelty of this manuscript.

Response: Thank you for the comment, and we agree the identity of CD24⁺ cells in the bone microenvironment are of critical interest. As also noted in the response to Reviewer 1, point 2, we have performed histological staining of CD24⁺ cells, demonstrating their co-expression with Runx2 in cells on the bone surface with a morphology consistent with osteoblasts and/or bone-lining cells (Extended Data Fig. 5C). Additionally, as noted above, we have performed CITE-Seq using the CD24 antibody, uncovering co-expression with Runx2 and Sp7 (Osterix), which indicate different osteolineage subpopulations (Extended Data Fig. 5D-F). Moreover, we performed differential expression analyses in our CITE-Seq data to define transcriptional profiles distinct in total CD24⁺ cells as well as CD24⁺ osteolineage cells (CD24⁺Runx2⁺Sp7⁺) vs CD24⁻ cells

(Extended Data Fig 5G, H). We found that CD24+ cells exhibited upregulated expression of osteolineage markers (*Bglap2*, *Bglap*, *Postn*, *Col1a1*), while progenitor markers (e.g., *Cxcl12*, *Lepr*, *Adipoq*) were more highly expressed in CD24- cells (Extended Data Fig. 5G). In CD24 osteolineage cells (CD24+*Runx2*+*Sp7*+), many osteogenic markers were upregulated (*Alpl*, *Bglap*, *Bglap2*), including multiple members of the collagen family (*Col1a1*, *Col11a1*, *Col12a1*, *Col13a1*), suggesting these cells may have a role in extracellular matrix production (Extended Data Fig. 5H). Like CD24+ cells, CD24 osteolineage cells were reduced in their expression of markers expressed by skeletal progenitors (*Cxcl12*²⁸ and *Ebf1*²⁹).

The study by van de Peppel et al. is important in supporting our findings, yet we believe the our manuscript is novel to the field due to (1) the identification of CD24+ osteolineage cells *in vivo* by CyTOF, scRNA-seq, and CITE-seq; (2) the detailed senescence phenotyping of CD24+ cells through multiparametric CyTOF analyses and *in vitro* SA- β -gal staining; and (3) the discovery that CD24+ cells are targeted by both genetic and pharmacologic senolytic therapies in aged mice. Our study thus provides evidence strongly linking CD24 to age-related skeletal senescence *in vivo*, which we believe builds upon previous *in vitro* studies on CD24, such as that by van de Peppel et al., that did not focus on senescence or aging.

Minor comments:

1. Can the author explain why D+Q cannot kill most of the p16KB cells. Why are the CD24+ cells so special?

Response: Although D+Q is now widely used as a senolytic cocktail in animal studies based on the original validation³⁰, its precise mechanism of action and specific cells targeted remain unclear. As such, we do not have a good explanation for why D+Q would kill CD24+ cells, although this may be related to their dependence on specific anti-apoptotic pathways targeted by D+Q, as originally described by Zhu et al.³⁰. This issue is unfortunately beyond the scope of the present studies, which largely focused on the more tractable genetic model (*INK-ATTAC*) and essentially used D+Q to further validate the genetic approach. In terms of the susceptibility of the CD24+ cells to AP20187 in the *INK-ATTAC* model, please see our response to Reviewer 1, point 1, where we show that *in vitro*, AP20187 causes apoptosis of CD24+, but not CD24-, cells from *INK-ATTAC* mice.

2. CD24 has become a very interesting marker in cancer and cancer immunotherapy. What may have stimulated CD24 expression in the osteogenic cells?

Response: Our *in vitro* data (Fig. 6H) indicate that CD24 is upregulated with the induction of senescence (along with p16 and p21), yet the mechanistic regulation of CD24 in osteolineage cells was beyond the scope of this project. We have added this point to the Discussion on page 31: “Additional *in vitro* and *in vivo* studies are needed, however, to further define the regulation of CD24 expression in senescent osteolineage cells as well as possible paracrine effects of these CD24 osteolineage cells on osteoblasts as well as osteoclasts and other cells in the bone microenvironment, including adipocytes and hematopoietic cells.”

Although speculative, JAK/STAT signaling may be involved, as STAT3 has been shown to upregulate CD24 expression in epithelial cells³¹ and regulate senescence in lung fibroblasts³². JAK/STAT3 signaling has been shown to regulate senescence in skeletal mesenchymal cells³³, and treatment of mice with JAK inhibitor ruxolitinib – which can specifically inactivate STAT3³⁴ – alleviated age-related bone loss⁵. CD24 may be upregulated due to this interaction, and thus our future studies are aimed at assessing the regulation of CD24 in osteolineage cells, which may also provide important links to its regulation in cancer cells.

3. I am not sure that etoposide is a proper inducer of p16 as it is a genotoxic drug that activates the p53 pathway.

Response: The Reviewer is correct, and we have generally seen upregulation of p21, which is downstream of p53, by etoposide. However, as shown in Fig.6I, we also observed upregulation of p16 by etoposide in BMSCs by 7 days post-treatment. Furthermore, our goal here was not necessarily to activate p16, but rather to test whether senescence can induce CD24 expression. We simply chose etoposide as it induces a robust senescence phenotype in BMSCs³⁵. We acknowledge, however, that further studies are needed to evaluate whether specifically p16-mediated senescence is associated with upregulation of CD24. This is now included in the Discussion on page 31 (also addresses point 2, above): “Additional *in vitro* and *in vivo* studies are needed, however, to further define the regulation of CD24 expression in senescent osteolineage cells as well as possible paracrine effects of these CD24 osteolineage cells on osteoblasts as well as osteoclasts and other cells in the bone microenvironment, including adipocytes and hematopoietic cells.”

References

1. Garcia-Sifuentes Y, Maney DL. Reporting and misreporting of sex differences in the biological sciences. *Elife*. 2021;10. Epub 2021/11/03. doi: 10.7554/eLife.70817. PubMed PMID: 34726154; PMCID: PMC8562995.
2. Baker DJ, Childs BG, Durik M, Wijers ME, Sieben CJ, Zhong J, Saltness RA, Jeganathan KB, Verzosa GC, Pezeshki A, Khazaie K, Miller JD, van Deursen JM. Naturally occurring p16(Ink4a)-positive cells shorten healthy lifespan. *Nature*. 2016;530(7589):184-9. Epub 2016/02/04. doi: 10.1038/nature16932. PubMed PMID: 26840489; PMCID: PMC4845101.
3. Baker DJ, Wijshake T, Tchkonja T, LeBrasseur NK, Childs BG, van de Sluis B, Kirkland JL, van Deursen JM. Clearance of p16Ink4a-positive senescent cells delays ageing-associated disorders. *Nature*. 2011;479(7372):232-6. Epub 2011/11/04. doi: 10.1038/nature10600. PubMed PMID: 22048312; PMCID: PMC3468323.
4. Ayre DC, Christian SL. CD24: A Rheostat That Modulates Cell Surface Receptor Signaling of Diverse Receptors. *Front Cell Dev Biol*. 2016;4:146. Epub 2017/01/14. doi: 10.3389/fcell.2016.00146. PubMed PMID: 28083532; PMCID: PMC5186806.
5. Farr JN, Xu M, Weivoda MM, Monroe DG, Fraser DG, Onken JL, Negley BA, Sfeir JG, Ogrodnik MB, Hachfeld CM, LeBrasseur NK, Drake MT, Pignolo RJ, Pirtskhalava T, Tchkonja T, Oursler MJ, Kirkland JL, Khosla S. Targeting cellular senescence prevents age-related bone loss in mice. *Nat Med*. 2017;23(9):1072-9. Epub 2017/08/22. doi: 10.1038/nm.4385. PubMed PMID: 28825716; PMCID: PMC5657592.
6. Wang JS, Kamath T, Mazur CM, Mirzamohammadi F, Rotter D, Hojo H, Castro CD, Tokavanich N, Patel R, Govea N, Enishi T, Wu Y, da Silva Martins J, Bruce M, Brooks DJ, Bouxsein ML, Tokarz D, Lin CP, Abdul A, Macosko EZ, Fiscoletti M, Munns CF, Ryder P, Kost-Alimova M, Byrne P, Cimini B, Fujiwara M, Kronenberg HM, Wein MN. Control of osteocyte dendrite formation by Sp7 and its target gene osteocrin. *Nat Commun*. 2021;12(1):6271. Epub 2021/11/03. doi: 10.1038/s41467-021-26571-7. PubMed PMID: 34725346; PMCID: PMC8560803.
7. Baryawno N, Przybylski D, Kowalczyk MS, Kfoury Y, Severe N, Gustafsson K, Kokkalis KD, Mercier F, Tabaka M, Hofree M, Dionne D, Papazian A, Lee D, Ashenberg O, Subramanian A, Vaishnav ED, Rozenblatt-Rosen O, Regev A, Scadden DT. A Cellular Taxonomy of the Bone Marrow Stroma in Homeostasis and Leukemia. *Cell*. 2019;177(7):1915-32 e16. Epub 2019/05/28. doi: 10.1016/j.cell.2019.04.040. PubMed PMID: 31130381; PMCID: PMC6570562.
8. Aquino-Martinez R, Monroe DG, Ventura F. Calcium mimics the chemotactic effect of conditioned media and is an effective inducer of bone regeneration. *PLoS One*. 2019;14(1):e0210301. Epub 2019/01/05. doi: 10.1371/journal.pone.0210301. PubMed PMID: 30608979; PMCID: PMC6319750.

9. Severe N, Karabacak NM, Gustafsson K, Baryawno N, Courties G, Kfoury Y, Kokkaliaris KD, Rhee C, Lee D, Scadden EW, Garcia-Robledo JE, Brouse T, Nahrendorf M, Toner M, Scadden DT. Stress-Induced Changes in Bone Marrow Stromal Cell Populations Revealed through Single-Cell Protein Expression Mapping. *Cell Stem Cell*. 2019;25(4):570-83 e7. Epub 2019/07/08. doi: 10.1016/j.stem.2019.06.003. PubMed PMID: 31279774; PMCID: PMC6778015.
10. Block MS, Dietz AB, Gustafson MP, Kalli KR, Erskine CL, Youssef B, Vijay GV, Allred JB, Pavelko KD, Strausbauch MA, Lin Y, Grudem ME, Jatoi A, Klampe CM, Wahner-Hendrickson AE, Weroha SJ, Glaser GE, Kumar A, Langstraat CL, Solseth ML, Deeds MC, Knutson KL, Cannon MJ. Th17-inducing autologous dendritic cell vaccination promotes antigen-specific cellular and humoral immunity in ovarian cancer patients. *Nat Commun*. 2020;11(1):5173. Epub 2020/10/16. doi: 10.1038/s41467-020-18962-z. PubMed PMID: 33057068; PMCID: PMC7560895 Folate Receptors', which is currently licensed to Marker Therapeutics, Inc. of Houston, TX. M.J.C. is an inventor on a patent filed by the University of Arkansas, entitled 'Inhibition of dendritic cell-driven regulatory T cell activation and potentiation of tumor antigen-specific T cell responses by interleukin-15 and MAP kinase inhibitor'. The remaining authors have no competing interests.
11. Zhang X, Pearsall VM, Carver CM, Atkinson EJ, Clarkson BDS, Grund EM, Baez-Faria M, Pavelko KD, Kachergus JM, White TA, Johnson RK, Malo CS, Gonzalez-Suarez AM, Ayasoufi K, Johnson KO, Tritz ZP, Fain CE, Khadka RH, Ogrodnik M, Jurk D, Zhu Y, Tchkonja T, Revzin A, Kirkland JL, Johnson AJ, Howe CL, Thompson EA, LeBrasseur NK, Schafer MJ. Rejuvenation of the aged brain immune cell landscape in mice through p16-positive senescent cell clearance. *Nat Commun*. 2022;13(1):5671. Epub 2022/09/28. doi: 10.1038/s41467-022-33226-8. PubMed PMID: 36167854; PMCID: PMC9515187 intellectual property related to this research. This research was reviewed by the Mayo Clinic Conflict of Interest Review Board and was conducted in compliance with Mayo Clinic Conflict of Interest policies. All other authors declare no competing interests.
12. Lagnado A, Leslie J, Ruchaud-Sparagano MH, Victorelli S, Hirsova P, Ogrodnik M, Collins AL, Vizioli MG, Habiballa L, Saretzki G, Evans SA, Salmonowicz H, Hruby A, Geh D, Pavelko KD, Dolan D, Reeves HL, Grellescheid S, Wilson CH, Pandanaboyana S, Doolittle M, von Zglinicki T, Oakley F, Gallage S, Wilson CL, Birch J, Carroll B, Chapman J, Heikenwalder M, Neretti N, Khosla S, Masuda CA, Tchkonja T, Kirkland JL, Jurk D, Mann DA, Passos JF. Neutrophils induce paracrine telomere dysfunction and senescence in ROS-dependent manner. *EMBO J*. 2021;40(9):e106048. Epub 2021/03/26. doi: 10.15252/embj.2020106048. PubMed PMID: 33764576; PMCID: PMC8090854.
13. Bendall SC, Simonds EF, Qiu P, Amir el AD, Krutzik PO, Finck R, Bruggner RV, Melamed R, Trejo A, Ornatsky OI, Balderas RS, Plevritis SK, Sachs K, Pe'er D, Tanner SD, Nolan GP. Single-cell mass cytometry of differential immune and drug responses across a human hematopoietic continuum. *Science*. 2011;332(6030):687-96. Epub 2011/05/10. doi: 10.1126/science.1198704. PubMed PMID: 21551058; PMCID: PMC3273988.
14. Nicolet BP, Guislain A, van Alphen FPJ, Gomez-Eerland R, Schumacher TNM, van den Biggelaar M, Wolkers MC. CD29 identifies IFN-gamma-producing human CD8(+) T cells with an increased cytotoxic potential. *Proc Natl Acad Sci U S A*. 2020;117(12):6686-96. Epub 2020/03/13. doi: 10.1073/pnas.1913940117. PubMed PMID: 32161126; PMCID: PMC7104308.
15. Ketola K, Vainio P, Fey V, Kallioniemi O, Iljin K. Monensin is a potent inducer of oxidative stress and inhibitor of androgen signaling leading to apoptosis in prostate cancer cells. *Mol Cancer Ther*. 2010;9(12):3175-85. Epub 2010/12/17. doi: 10.1158/1535-7163.MCT-10-0368. PubMed PMID: 21159605.
16. Charvat RA, Arrizabalaga G. Oxidative stress generated during monensin treatment contributes to altered *Toxoplasma gondii* mitochondrial function. *Sci Rep*. 2016;6:22997. Epub 2016/03/16. doi: 10.1038/srep22997. PubMed PMID: 26976749; PMCID: PMC4792157.
17. Nylander S, Kalies I, Brefeldin A, but not monensin, completely blocks CD69 expression on mouse lymphocytes: efficacy of inhibitors of protein secretion in protocols for intracellular cytokine

staining by flow cytometry. *J Immunol Methods*. 1999;224(1-2):69-76. Epub 1999/06/05. doi: 10.1016/s0022-1759(99)00010-1. PubMed PMID: 10357208.

18. Reyes NS, Krasilnikov M, Allen NC, Lee JY, Hyams B, Zhou M, Ravishankar S, Cassandras M, Wang C, Khan I, Matatia P, Johmura Y, Molofsky A, Matthay M, Nakanishi M, Sheppard D, Campisi J, Peng T. Sentinel p16(INK4a+) cells in the basement membrane form a reparative niche in the lung. *Science*. 2022;378(6616):192-201. Epub 2022/10/14. doi: 10.1126/science.abf3326. PubMed PMID: 36227993.

19. Hartman ML, Czyz M. BCL-w: apoptotic and non-apoptotic role in health and disease. *Cell Death Dis*. 2020;11(4):260. Epub 2020/04/23. doi: 10.1038/s41419-020-2417-0. PubMed PMID: 32317622; PMCID: PMC7174325.

20. Kozopas KM, Yang T, Buchan HL, Zhou P, Craig RW. MCL1, a gene expressed in programmed myeloid cell differentiation, has sequence similarity to BCL2. *Proc Natl Acad Sci U S A*. 1993;90(8):3516-20. Epub 1993/04/15. doi: 10.1073/pnas.90.8.3516. PubMed PMID: 7682708; PMCID: PMC46331.

21. Troiani M, Colucci M, D'Ambrosio M, Guccini I, Pasquini E, Varesi A, Valdata A, Mosole S, Revandkar A, Attanasio G, Rinaldi A, Rinaldi A, Bolis M, Cippa P, Alimonti A. Single-cell transcriptomics identifies Mcl-1 as a target for senolytic therapy in cancer. *Nat Commun*. 2022;13(1):2177. Epub 2022/04/23. doi: 10.1038/s41467-022-29824-1. PubMed PMID: 35449130; PMCID: PMC9023465 inventors of the patent WO2019142095A1 (Title: new alk inhibitor senolytic drugs). The remaining authors declare no competing interests.

22. Deveraux QL, Takahashi R, Salvesen GS, Reed JC. X-linked IAP is a direct inhibitor of cell-death proteases. *Nature*. 1997;388(6639):300-4. Epub 1997/07/17. doi: 10.1038/40901. PubMed PMID: 9230442.

23. Ren J, Shi M, Liu R, Yang QH, Johnson T, Skarnes WC, Du C. The Birc6 (Bruce) gene regulates p53 and the mitochondrial pathway of apoptosis and is essential for mouse embryonic development. *Proc Natl Acad Sci U S A*. 2005;102(3):565-70. Epub 2005/01/11. doi: 10.1073/pnas.0408744102. PubMed PMID: 15640352; PMCID: PMC543482.

24. Slobodnyuk K, Radic N, Ivanova S, Llado A, Trempolec N, Zorzano A, Nebreda AR. Autophagy-induced senescence is regulated by p38alpha signaling. *Cell Death Dis*. 2019;10(6):376. Epub 2019/05/17. doi: 10.1038/s41419-019-1607-0. PubMed PMID: 31092814; PMCID: PMC6520338.

25. Weng PW, Pikatan NW, Setiawan SA, Yadav VK, Fong IH, Hsu CH, Yeh CT, Lee WH. Role of GDF15/MAPK14 Axis in Chondrocyte Senescence as a Novel Senomorphic Agent in Osteoarthritis. *Int J Mol Sci*. 2022;23(13). Epub 2022/07/10. doi: 10.3390/ijms23137043. PubMed PMID: 35806043; PMCID: PMC9266723.

26. Freund A, Patil CK, Campisi J. p38MAPK is a novel DNA damage response-independent regulator of the senescence-associated secretory phenotype. *EMBO J*. 2011;30(8):1536-48. Epub 2011/03/15. doi: 10.1038/emboj.2011.69. PubMed PMID: 21399611; PMCID: PMC3102277.

27. Saul D, Kosinsky RL, Atkinson EJ, Doolittle ML, Zhang X, LeBrasseur NK, Pignolo RJ, Robbins PD, Niedernhofer LJ, Ikeno Y, Jurk D, Passos JF, Hickson LJ, Xue A, Monroe DG, Tchkonja T, Kirkland JL, Farr JN, Khosla S. A new gene set identifies senescent cells and predicts senescence-associated pathways across tissues. *Nat Commun*. 2022;13(1):4827. Epub 2022/08/17. doi: 10.1038/s41467-022-32552-1. PubMed PMID: 35974106; PMCID: PMC9381717.

28. Greenbaum A, Hsu YM, Day RB, Schuettpelz LG, Christopher MJ, Borgerding JN, Nagasawa T, Link DC. CXCL12 in early mesenchymal progenitors is required for haematopoietic stem-cell maintenance. *Nature*. 2013;495(7440):227-30. Epub 2013/02/26. doi: 10.1038/nature11926. PubMed PMID: 23434756; PMCID: PMC3600148.

29. Derecka M, Herman JS, Cauchy P, Ramamoorthy S, Lupar E, Grun D, Grosschedl R. EBF1-deficient bone marrow stroma elicits persistent changes in HSC potential. *Nat Immunol*.

- 2020;21(3):261-73. Epub 2020/02/19. doi: 10.1038/s41590-020-0595-7. PubMed PMID: 32066955.
30. Zhu Y, Tchkonina T, Pirtskhalava T, Gower AC, Ding H, Giorgadze N, Palmer AK, Ikeno Y, Hubbard GB, Lenburg M, O'Hara SP, LaRusso NF, Miller JD, Roos CM, Verzosa GC, LeBrasseur NK, Wren JD, Farr JN, Khosla S, Stout MB, McGowan SJ, Fuhrmann-Stroissnigg H, Gurkar AU, Zhao J, Colangelo D, Dorransoro A, Ling YY, Barghouthy AS, Navarro DC, Sano T, Robbins PD, Niedernhofer LJ, Kirkland JL. The Achilles' heel of senescent cells: from transcriptome to senolytic drugs. *Aging Cell*. 2015;14(4):644-58. Epub 2015/03/11. doi: 10.1111/ace.12344. PubMed PMID: 25754370; PMCID: PMC4531078.
31. Huser L, Sachindra S, Granados K, Federico A, Larribere L, Novak D, Umansky V, Altevogt P, Utikal J. SOX2-mediated upregulation of CD24 promotes adaptive resistance toward targeted therapy in melanoma. *Int J Cancer*. 2018;143(12):3131-42. Epub 2018/06/16. doi: 10.1002/ijc.31609. PubMed PMID: 29905375.
32. Waters DW, Blokland KEC, Pathinayake PS, Wei L, Schuliga M, Jaffar J, Westall GP, Hansbro PM, Prele CM, Mutsaers SE, Bartlett NW, Burgess JK, Grainge CL, Knight DA. STAT3 Regulates the Onset of Oxidant-induced Senescence in Lung Fibroblasts. *Am J Respir Cell Mol Biol*. 2019;61(1):61-73. Epub 2019/01/05. doi: 10.1165/rcmb.2018-0328OC. PubMed PMID: 30608861.
33. Wu W, Fu J, Gu Y, Wei Y, Ma P, Wu J. JAK2/STAT3 regulates estrogen-related senescence of bone marrow stem cells. *J Endocrinol*. 2020;245(1):141-53. Epub 2020/02/12. doi: 10.1530/JOE-19-0518. PubMed PMID: 32045363.
34. Qureshy Z, Li H, Zeng Y, Rivera J, Cheng N, Peterson CN, Kim MO, Ryan WR, Ha PK, Bauman JE, Wang SJ, Long SR, Johnson DE, Grandis JR. STAT3 Activation as a Predictive Biomarker for Ruxolitinib Response in Head and Neck Cancer. *Clin Cancer Res*. 2022;28(21):4737-46. Epub 2022/08/06. doi: 10.1158/1078-0432.CCR-22-0744. PubMed PMID: 35929989.
35. Kaur J, Saul D, Doolittle ML, Rowsey JL, Vos SJ, Farr JN, Khosla S, Monroe DG. Identification of a suitable endogenous control miRNA in bone aging and senescence. *Gene*. 2022;835:146642. Epub 2022/06/15. doi: 10.1016/j.gene.2022.146642. PubMed PMID: 35700807; PMCID: PMC9533812.

REVIEWER COMMENTS

Reviewer #1 (Remarks to the Author):

The authors submitted the revised manuscript in a relatively short time. The manuscript is accompanied by a point-by-point response letter addressing the concerns raised by three reviewers. To support the major conclusion of the study, the authors performed additional analyses including CD24 immunohistochemistry and CITE-Seq. The authors' responses are mostly reasonable. Below are my comments on the authors' responses to my queries.

1. The relationship between p16 and CD24+ cells. The newly added plots of p16 and FLAG (the readout of INK-ATTAC expression) are extremely important to interpret how genetic senolysis acts on CD24+ cells. Please move these two plots (currently Extended Data Figure 8 D-F) to the main Figure 3, showing side-by-side with the CD24 Vehicle/AP plots, in panel F.

2. The identity of CD24+ cells. This corroborates with the other reviewers' major points (Reviewer 2: Point 4, Reviewer 3, Point 2). Despite additional data, the identity of CD24+ cells still remains largely mysterious. Newly included Extended Data Figure 5C (immunohistochemistry) shows that CD24 is broadly expressed by marrow hematopoietic and stromal cells, confirming the highly heterogeneous nature of CD24+ cells. Importantly, CD24 does not appear to be expressed by osteocytes. It is unclear whether CD24 marks a functionally distinct cell population that is particularly susceptible to senolysis. CD24 CITE-Seq is exciting, but it does not show a segregation of CD24+Runx2+Sp7+ and CD24+Sp7+ cells as the authors claim. It seems obvious that CD24+Sp7+ represents a subset of CD24+Runx2+Sp7+ cells. The CITE-Seq data only highlight discordance between proteomics and transcriptomics-based definition of cell populations. A more likely scenario is that a subset of CD24+Runx2+Sp7+ cells undergo post-translational degradation of RUNX2, and become CD24+Sp7+. This makes sense because RUNX2 is known to inhibit terminal differentiation of osteoblasts. It is possible that genetic senolysis preferentially removes cells of the osteoblast lineage at the terminal stage of differentiation, regardless of CD24 status. This can explain why late osteoblasts/osteocytes lacking CD24 were removed by AP. I am afraid that the authors emphasize too much the novelty regarding CD24+ cells as a target of senolysis. My recommendation is to change the statement in the title and abstract from "CD24 osteolineage cells" to "CD24high osteolineage cells and late osteoblasts/cytes" to accurately present the findings.

3. CD24 and INKBRITE. The authors' response is acceptable.

4. Senolysis-induced apoptosis of CD24+ cells. The authors showed that Annexin V+ apoptotic cells increased by AP in vitro. Although statistically significant, the difference was modest (from 1.0% to 1.25%). This leaves the possibility that other mechanisms other than apoptosis might be at work, including the scenario that I mentioned (i.e. blockade of terminal differentiation of early osteoblasts). The authors should critically appraise the limitation of their current supporting data, and discuss alternative possibilities in the revised manuscript.

5. Cell culture of CD24+ cells: The response is acceptable.

12. Discordance of CyTOF and scRNA-seq is also highlighted by the authors' responses regarding Sost/Sclerostin.

13. The authors should acknowledge alternative possibilities that apoptosis may not be the only mechanism supporting the disappearance of CD24+ cells.

Additional individual points in the revised manuscript:

1. Line 30: The authors' data clearly demonstrate that not all CD24+ osteolineage cells are cleared by genetic senolytic clearance. It should be changed to "CD24^{high} osteolineage cells and late osteoblasts/cytes" for accuracy.
2. Line 262: The authors should state that CD24 signals were broadly observed in bone marrow, but not in osteocytes, on sections.
3. Line 267: CITE-Seq does not show a segregation of CD24+Runx2+Sp7+ and CD24+Sp7+ cells. According to Extended Data Figure 5D,E, CD24+Sp7+ is a subset of CD24+Runx2+Sp7+, suggesting a more interesting explanation regarding post-transcriptional regulation of RUNX2.
4. Line 349: The authors should also mention alternative scenarios of CD24+ cell clearance, considering modest changes.

Reviewer #2 (Remarks to the Author):

The authors have covered most of the comments included by the reviewers. I think this version of the paper is much clearer and better written than the last. I have a few minor comments to be addressed which I have written below:

Minor Comments:

-Extended data 5B and Pg14 line 254-255. "Note that the latter cells emerged as a distinct cluster and had lower expression of Runx2 as well as somewhat lower expression of Osterix than the CD24^{high}/low osteolineage (CD24+/Runx2+/Osterix+) cells (Fig 3B)". Although the CD24+/Osterix+ cluster overall as a bulk population has lower Osterix expression (Fig 3B); a sub-population of this cluster has extremely high Osterix expression on par if not higher than the CD24^{high}/low clusters (Extended fig 5B). Could the authors rewrite this statement to point this out? This sub-population is clearly distinct from the population as a whole.

-The wrong figure numbers are written in the text making it a little confusing. For example, pg 11 lines 200-203 extended data fig 3D, 3E, 3F, Pg 15 line 286, 288.

Reviewer #3 (Remarks to the Author):

The authors have addressed most of the concerns although the identify of the CD24+ skeletal cells remains unclear. While the authors believe that CD24+ cells are differentiated osteoblasts, their cell surface marker analysis supports that these cells are progenitors, which is supported by a previous study (van de Peppel, J. et al. Cell Surface Glycoprotein CD24 Marks Bone Marrow-Derived Human Mesenchymal Stem/Stromal Cells with Reduced Proliferative and Differentiation Capacity In Vitro. Stem Cells Dev 30, 325-336 (2021).<https://doi.org/10.1089/scd.2021.0027>).

Response to Reviewers

Note: All changes to the previous submission are indicated by tracking and page numbers below refer to the tracked version. We are also including a “clean” version of the manuscript.

Reviewer #1

The authors submitted the revised manuscript in a relatively short time. The manuscript is accompanied by a point-by-point response letter addressing the concerns raised by three reviewers. To support the major conclusion of the study, the authors performed additional analyses including CD24 immunohistochemistry and CITE-Seq. The authors' responses are mostly reasonable. Below are my comments on the authors' responses to my queries.

Response: We thank the Reviewer for the positive comments regarding our revisions.

1. The relationship between p16 and CD24+ cells. The newly added plots of p16 and FLAG (the readout of INK-ATTAC expression) are extremely important to interpret how genetic senolysis acts on CD24+ cells. Please move these two plots (currently Extended Data Figure 8 D-F) to the main Figure 3, showing side-by-side with the CD24 Vehicle/AP plots, in panel F.

Response: We agree and have moved these plots to main Figure 3I directly underneath the CD24 Vehicle/AP plots in Figure 3H.

2. The identity of CD24+ cells. This corroborates with the other reviewers' major points (Reviewer 2: Point 4, Reviewer 3, Point 2). Despite additional data, the identity of CD24+ cells still remains largely mysterious. Newly included Extended Data Figure 5C (immunohistochemistry) shows that CD24 is broadly expressed by marrow hematopoietic and stromal cells, confirming the highly heterogenous nature of CD24+ cells. Importantly, CD24 does not appear to be expressed by osteocytes.

Response: These cellular annotations on the IHC data have been incorporated into the Results on page 14, where we specifically note: “CD24 staining was also evident in marrow cells lacking Runx2 expression, likely B cells⁷⁹, but CD24 was not expressed in osteocytes.”

It is unclear whether CD24 marks a functionally distinct cell population that is particularly susceptible to senolysis. CD24 CITE-Seq is exciting, but it does not show a segregation of CD24+Runx2+Sp7+ and CD24+Sp7+ cells as the authors claim. It seems obvious that CD24+Sp7+ represents a subset of CD24+Runx2+Sp7+ cells. The CITE-Seq data only highlight discordance between proteomics and transcriptomics-based definition of cell populations. A more likely scenario is that a subset of CD24+Runx2+Sp7+ cells undergo post-translational degradation of RUNX2, and become CD24+Sp7+. This makes sense because RUNX2 is known to inhibit terminal differentiation of osteoblasts. It is possible that genetic senolysis preferentially removes cells of the osteoblast lineage at the terminal stage of differentiation, regardless of CD24 status. This can explain why late osteoblasts/osteocytes lacking CD24 were removed by AP. I am afraid that the authors emphasize too much the novelty regarding CD24+ cells as a target of senolysis. My recommendation is to change the statement in the title and abstract from “CD24 osteolineage cells” to “CD24^{high} osteolineage cells and late osteoblasts/cytes” to accurately present the findings

Response: We agree, and the Reviewer makes reasonable points. We have changed the title of the paper as suggested and also substantially modified the abstract along the lines suggested by the Reviewer. With regards to the additional issues regarding the CD24+ cells noted by the Reviewer, we now explicitly address the issue of possible post-translational modification of RUNX2 on page 15 in the context of the interpretation of the CITE-Seq data: “There was less segregation between these populations than what was observed at the protein level by CyTOF,

which suggests possible post-translational degradation of RUNX2 that generates a distinct CD24+Osterix+ population.” To address the other point of the Reviewer, we note in the Discussion on page 33: “Along these lines, we cannot exclude the possibility that genetic senolysis preferentially removes cells of the osteoblast lineage at the terminal stage of differentiation, regardless of CD24 status, resulting in elimination of late osteoblastic cells expressing CD24 (although osteocytes do not appear to express CD24, Extended Data Fig. 5C). Thus, additional *in vitro* and *in vivo* studies are needed to further define the regulation of CD24 expression in senescent osteolineage cells as well as possible paracrine effects of these CD24 osteolineage cells on osteoblasts as well as osteoclasts and other cells in the bone microenvironment, including adipocytes and hematopoietic cells.”

3. *CD24 and INKBRITE. The authors’ response is acceptable.*

Response: We thank the Reviewer for the acceptance of our response.

4. *Senolysis-induced apoptosis of CD24+ cells. The authors showed that Annexin V+ apoptotic cells increased by AP in vitro. Although statistically significant, the difference was modest (from 1.0% to 1.25%). This leaves the possibility that other mechanisms other than apoptosis might be at work, including the scenario that I mentioned (i.e. blockade of terminal differentiation of early osteoblasts). The authors should critically appraise the limitation of their current supporting data, and discuss alternative possibilities in the revised manuscript.*

Response: We have further expanded on these issues, as suggested by the Reviewer. Part of our response to this point includes our response to point 2, above, where we acknowledge alternate explanations for the senolytic clearance of CD24+ cells. We have further addressed this limitation in the Results on page 18 as follows: “However, despite these findings demonstrating the pro-apoptotic effects of AP20187 on cell populations enriched for p16+ cells, it is possible the robust *in vivo* reduction of late osteoblastic/osteocytic cells observed by CyTOF may also be driven by additional non-apoptotic routes, including that treatment with AP20187 resulted in blockade of terminal differentiation of early osteoblasts, and additional studies are needed to address this possibility.”

5. *Cell culture of CD24+ cells: The response is acceptable.*

Response: We thank the Reviewer for the acceptance of our response.

12. *Discordance of CyTOF and scRNA-seq is also highlighted by the authors’ responses regarding Sost/Sclerostin.*

Response: We thank the Reviewer.

13. *The authors should acknowledge alternative possibilities that apoptosis may not be the only mechanism supporting the disappearance of CD24+ cells.*

Response: We have acknowledged these alternative possibilities in the Results as stated above in Points 2 and 4.

Additional individual points in the revised manuscript:

1. *Line 30: The authors’ data clearly demonstrate that not all CD24+ osteolineage cells are cleared by genetic senolytic clearance. It should be changed to “CD24high osteolineage cells and late osteoblasts/cytes” for accuracy.*

Response: We agree and have addressed this issue in the title, abstract, and throughout the manuscript.

2. *Line 262: The authors should state that CD24 signals were broadly observed in bone marrow, but not in osteocytes, on sections.*

Response: We have made the appropriate changes in the Results (page 14).

3. Line 267: *CITE-Seq does not show a segregation of CD24+Runx2+Sp7+ and CD24+Sp7+ cells. According to Extended Data Figure 5D,E, CD24+Sp7+ is a subset of CD24+Runx2+Sp7+, suggesting a more interesting explanation regarding post-transcriptional regulation of RUNX2.*

Response: We have changed our interpretation in the Results, including the possibility for post-transcriptional degradation of Runx2 (see response to point 2, above).

4. Line 349: *The authors should also mention alternative scenarios of CD24+ cell clearance, considering modest changes.*

Response: We have made the appropriate changes, as noted in the responses above.

Reviewer #2:

The authors have covered most of the comments included by the reviewers. I think this version of the paper is much clearer and better written than the last. I have a few minor comments to be addressed which I have written below:

Response: We thank the Reviewer for the positive comments regarding our revisions.

Minor Comments:

-Extended data 5B and Pg14 line 254-255. "Note that the latter cells emerged as a distinct cluster and had lower expression of Runx2 as well as somewhat lower expression of Osterix than the CD24^{high/low} osteolineage (CD24+/Runx2+/Osterix+) cells (Fig 3B)". Although the CD24+/Osterix+ cluster overall as a bulk population has lower Osterix expression (Fig 3B); a sub-population of this cluster has extremely high Osterix expression on par if not higher than the CD24^{high/low} clusters (Extended fig 5B). Could the authors rewrite this statement to point this out? This sub-population is clearly distinct from the population as a whole.

Response: We agree with the Reviewer, and have rewritten as follows on page 14: "Note that the latter cells emerged as a distinct cluster and had lower expression of Runx2 than the CD24^{high/low} osteolineage (CD24+/Runx2+/Osterix+) cells (Fig. 3B), yet had strong Osterix positivity within a large portion of this cluster (Extended Data Fig. 5A, B).

-The wrong figure numbers are written in the text making it a little confusing. For example, pg 11 lines 200-203 extended data fig 3D, 3E, 3F, Pg 15 line 286, 288.

Response: Our apologies, we have gone through and fixed figure numbers appropriately.

Reviewer #3:

The authors have addressed most of the concerns although the identify of the CD24+ skeletal cells remains unclear. While the authors believe that CD24+ cells are differentiated osteoblasts, their cell surface marker analysis supports that these cells are progenitors, which is supported by a previous study (van de Peppel, J. et al. Cell Surface Glycoprotein CD24 Marks Bone Marrow-Derived Human Mesenchymal Stem/Stromal Cells with Reduced Proliferative and Differentiation Capacity In Vitro. Stem Cells Dev 30, 325-336 (2021). 10.1089/scd.2021.0027).

Response: We agree that, at this point, we are unable to definitively state at which point of osteogenesis the CD24+ cells lie. We have made this point in the Discussion on pages 32-33 as follows: "Notably, as revealed by our CITE-Seq analyses, the expression of early (*Runx2*), mid (*Sp7*), and late (*Bglap*) osteogenic genes within CD24+ cells clouds whether these cells are

progenitors or late osteogenic cells, or perhaps a heterogeneous mix of cells at various stages of osteoblastic differentiation.”

REVIEWERS' COMMENTS

Reviewer #1 (Remarks to the Author):

I would like to congratulate the authors on this excellent study. This reviewer has no further comments.